# How biological attention mechanisms improve task performance in a large-scale visual system model

Grace W Lindsay[1,2]*, Kenneth D Miller[1,2,3,4]

[1]Center for Theoretical Neuroscience, College of Physicians and Surgeons, Columbia University, New York, United States; [2]Mortimer B. Zuckerman Mind Brain Behaviour Institute, Columbia University, New York, United States; [3]Swartz Program in Theoretical Neuroscience, Kavli Institute for Brain Science, New York, United States; [4]Department of Neuroscience, Columbia University, New York, United States

**Abstract** How does attentional modulation of neural activity enhance performance? Here we use a deep convolutional neural network as a large-scale model of the visual system to address this question. We model the feature similarity gain model of attention, in which attentional modulation is applied according to neural stimulus tuning. Using a variety of visual tasks, we show that neural modulations of the kind and magnitude observed experimentally lead to performance changes of the kind and magnitude observed experimentally. We find that, at earlier layers, attention applied according to tuning does not successfully propagate through the network, and has a weaker impact on performance than attention applied according to values computed for optimally modulating higher areas. This raises the question of whether biological attention might be applied at least in part to optimize function rather than strictly according to tuning. We suggest a simple experiment to distinguish these alternatives.

DOI: https://doi.org/10.7554/eLife.38105.001

*For correspondence:
gracewlindsay@gmail.com

**Competing interests:** The authors declare that no competing interests exist.

## Introduction

Covert visual attention—applied according to spatial location or visual features—has been shown repeatedly to enhance performance on challenging visual tasks (*Carrasco, 2011*). To explore the neural mechanisms behind this enhancement, neural responses to the same visual input are compared under different task conditions. Such experiments have identified numerous neural modulations associated with attention, including changes in firing rates, noise levels, and correlated activity (*Treue, 2001*; *Cohen and Maunsell, 2009*; *Fries et al., 2001*; *Maunsell and Cook, 2002*). But how do these neural activity changes impact performance? Previous theoretical studies have offered helpful insights on how attention may work to enhance performance (*Navalpakkam and Itti, 2007*; *Rolls and Deco, 2006*; *Tsotsos et al., 1995*; *Cave, 1999*; *Hamker and Worcester, 2002*; *Wolfe, 1994*; *Hamker, 1999*; *Eckstein et al., 2009*; *Borji and Itti, 2014*; *Whiteley and Sahani, 2012*; *Bundesen, 1990*; *Treisman and Gelade, 1980*; *Verghese, 2001*; *Chikkerur et al., 2010*). However, much of this work is either based on small, hand-designed models or lacks direct mechanistic interpretability. Here, we utilize a large-scale model of the ventral visual stream to explore the extent to which neural changes like those observed experimentally can lead to performance enhancements on realistic visual tasks. Specifically, we use a deep convolutional neural network trained to perform object classification to test effects of the feature similarity gain model of attention (*Treue and Martínez Trujillo, 1999*).

**eLife digest** Imagine you have lost your cell phone. Your eyes scan the cluttered table in front of you, searching for its familiar blue case. But what is happening within the visual areas of your brain while you search? One possibility is that neurons that represent relevant features such as 'blue' and 'rectangular' increase their activity. This might help you spot your phone among all the other objects on the table.

Paying attention to specific features improves our performance on visual tasks that require detecting those features. The 'feature similarity gain model' proposes that this is because attention increases the activity of neurons sensitive to specific target features, such as 'blue' in the example above. But is this how the brain solves such challenges in practice? Previous studies examining this issue have relied on correlations. They have shown that increases in neural activity correlate with improved performance on visual tasks. But correlation does not imply causation.

Lindsay and Miller have now used a computer model of the brain's visual pathway to examine whether changes in neural activity cause improved performance. The model was trained to use feature similarity gain to detect an object within a set of photographs. As predicted, changes in activity like those that occur in the brain did indeed improve the model's performance. Moreover, activity changes at later stages of the model's processing pathway produced bigger improvements than activity changes earlier in the pathway. This may explain why attention affects neural activity more at later stages in the visual pathway.

But feature similarity gain is not the only possible explanation for the results. Lindsay and Miller show that another pattern of activity change also enhanced the model's performance, and propose an experiment to distinguish between the two possibilities. Overall, these findings increase our understanding of how the brain processes sensory information. Work is ongoing to teach computers to process images as efficiently as the human visual system. The computer model used in this study is similar to those used in state-of-the-art computer vision. These findings could thus help advance artificial sensory processing too.

DOI: https://doi.org/10.7554/eLife.38105.002

Deep convolutional neural networks (CNNs) are popular tools in the machine learning and computer vision communities for performing challenging visual tasks (*Rawat and Wang, 2017*). Their architecture—comprised of layers of convolutions, nonlinearities, and response pooling—was designed to mimic the retinotopic and hierarchical nature of the mammalian visual system (*Rawat and Wang, 2017*). Models of a similar form have been used to study the biological underpinnings of object recognition for decades (*Fukushima, 1988*; *Riesenhuber and Poggio, 1999*; *Serre et al., 2007*). Recently it has been shown that when these networks are trained to successfully perform object classification on real-world images, the intermediate representations learned are remarkably similar to those of the primate visual system, making CNNs state-of-the-art models of the ventral stream (*Yamins et al., 2014*; *Khaligh-Razavi et al., 2017*; *Khaligh-Razavi and Kriege-skorte, 2014*; *Kheradpisheh et al., 2016*; *Kar et al., 2017*; *Cadena et al., 2017*; *Tripp, 2017*; *Love et al., 2017*; *Kubilius et al., 2016*). A key finding has been the correspondence between different areas in the ventral stream and layers in the deep CNNs, with early convolutional layers best able to capture the representation of V1 and middle and higher layers best able to capture V4 and IT, respectively (*Güçlü and van Gerven, 2015*; *Eickenberg et al., 2017*; *Yamins et al., 2014*). The generalizability of these networks is limited, however, and the models are not able to match all elements of visual behaviour (*Ullman et al., 2016*; *Azulay and Weiss, 2018*; *Baker et al., 2018*). But given that CNNs can reach near-human performance on some visual tasks and have architectural and representational similarities to the visual system, they are well-positioned for exploring how neural correlates of attention can impact behaviour.

One popular framework to describe attention's effects on firing rates is the feature similarity gain model (FSGM). This model, introduced by Treue and Martinez-Trujillo, claims that a neuron's activity is multiplicatively scaled up (or down) according to how much it prefers (or doesn't prefer) the properties of the attended stimulus (*Treue and Martínez Trujillo, 1999*; *Martinez-Trujillo and Treue, 2004*). Attention to a certain visual attribute, such as a specific orientation or color, is generally

referred to as feature-based attention (FBA). FBA effects are spatially global: if a task performed at one location in the visual field activates attention to a particular feature, neurons that represent that feature across the visual field will be affected (*Zhang and Luck, 2009*; *Saenz et al., 2002*). Overall, this leads to a general shift in the representation of the neural population towards that of the attended stimulus (*Çukur et al., 2013*; *Kaiser et al., 2016*; *Peelen and Kastner, 2011*). Spatial attention implies that a particular portion of the visual field is being attended. According to the FSGM, spatial location is treated as an attribute like any other. Therefore, a neuron's modulation due to attention can be predicted by how well it's preferred features and spatial receptive field align with the features or location of the attended stimulus. The effects of combined feature and spatial attention have been found to be additive (*Hayden and Gallant, 2009*).

A debated issue in the attention literature is where in the visual stream attention effects can be seen. Many studies of attention focus on V4 and MT/MST (*Treue, 2001*), as these areas have reliable attentional effects. Some studies do find effects at earlier areas (*Moro et al., 2010*), though they tend to be weaker and occur later in the visual response (*Kastner and Pinsk, 2004*). Therefore, a leading hypothesis is that attention signals, coming from prefrontal areas (*Moore and Armstrong, 2003*; *Monosov et al., 2011*; *Bichot et al., 2015*; *Kornblith and Tsao, 2017*), target later visual areas, and the feedback connections that those areas send to earlier ones cause the weaker effects seen there later (*Buffalo et al., 2010*; *Luck et al., 1997*).

In this study, we define the FSGM of attention mathematically and implement it in a deep CNN. By applying attention at different layers in the network and for different tasks, we see how neural changes at one area propagate through the network and change performance.

## Results

The network used in this study—VGG-16, (*Simonyan and Zisserman, 2014*)—is shown in *Figure 1A* and explained in Materials and methods, 'Network Model'. Briefly, at each convolutional layer, the application of a given convolutional filter results in a feature map, which is a 2-D grid of artificial neurons that represent how well the bottom-up input at each location aligns with the filter. Each layer has multiple feature maps. Therefore a 'retinotopic' layout is built into the structure of the network, and the same visual features are represented across that retinotopy (akin to how cells that prefer a given orientation exist at all locations across the V1 retinotopy). This network was explored in (*Güçlü and van Gerven, 2015*), where it was shown that early convolutional layers of this CNN are best at predicting activity of voxels in V1, while late convolutional layers are best at predicting activity of voxels in the object-selective lateral occipital area (LO).

### The relationship between tuning and classification

The feature similarity gain model of attention posits that neural activity is modulated by attention in proportion to how strongly a neuron prefers the attended features, as assessed by its tuning. However, the relationship between a neuron's tuning and its ability to influence downstream readouts remains a difficult one to investigate biologically. We use our hierarchical model to explore this question. We do so by using back propagation to calculate 'gradient values', which we compare to tuning curves (see Materials and methods, 'Object category gradient calculations' and 'Tuning values' for details). Gradient values indicate the ways in which feature map activities should change in order to make the network more likely to classify an image as being of a certain object category. Tuning values represent the degree to which the feature map responds preferentially to images of a given category. If there is a correspondence between tuning and classification, a feature map that prefers a given object category (that is, responds strongly to it) should also have a high positive gradient value for that category. In *Figure 2A* we show gradient values and tuning curves for three example feature maps. In *Figure 2C*, we show the average correlation coefficients between tuning values and gradient values for all feature maps at each of the 13 convolutional layers. As can be seen, tuning curves in all layers show higher correlation with gradient values than expected by chance (as assayed by shuffled controls), but this correlation is relatively low, increasing across layers from about .2 to .5. Overall tuning quality also increases with layer depth (*Figure 2B*), but less strongly.

Even at the highest layers, there can be serious discrepancies between tuning and gradient values. In *Figure 2D*, we show the gradient values of feature maps at the final four convolutional layers,

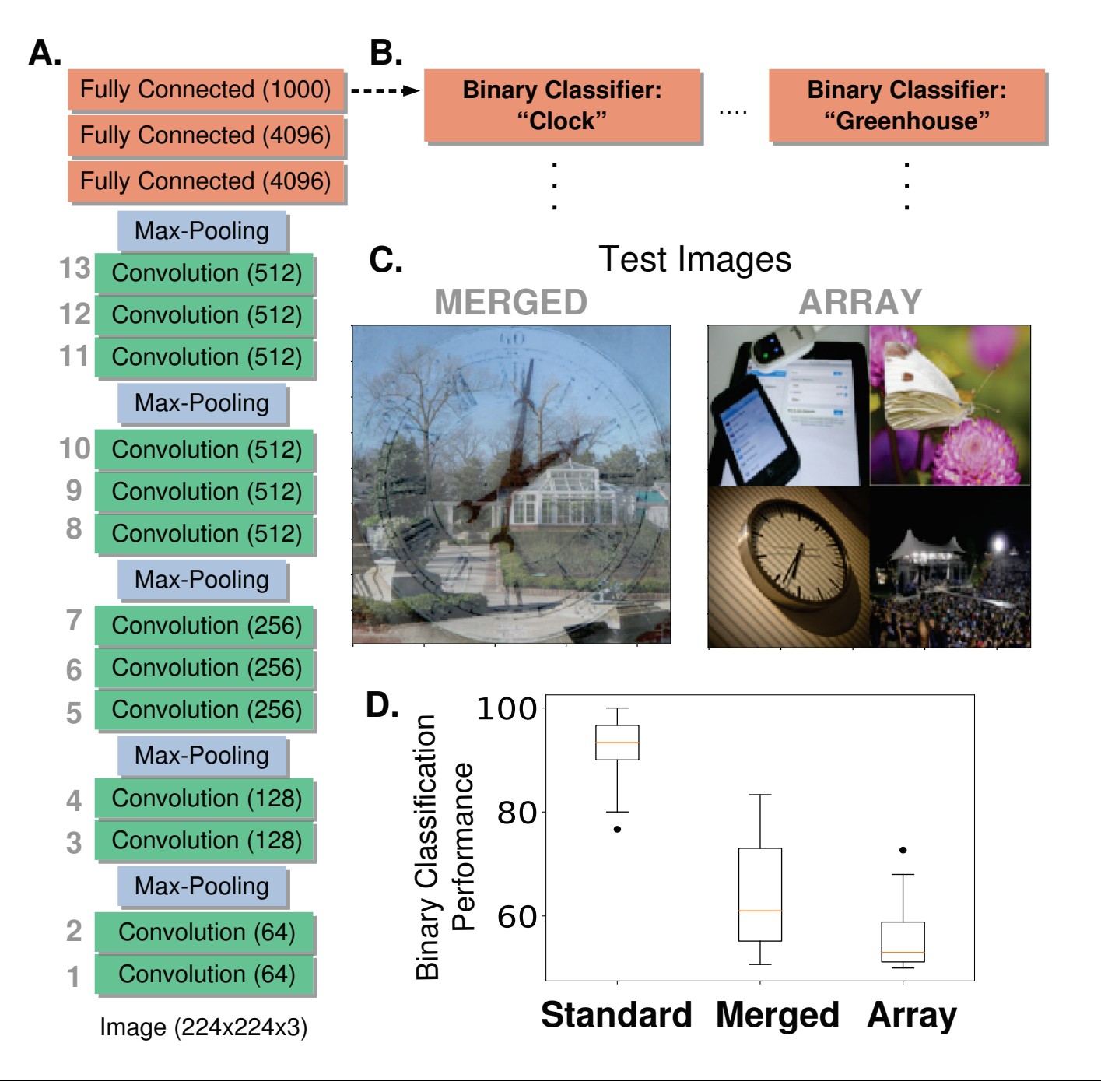

**Figure 1.** Network architecture and feature-based attention task setup. (**A**) The model used is a pre-trained deep neural network (VGG-16) that contains 13 convolutional layers (labelled in gray, number of feature maps given in parenthesis) and is trained on the ImageNet dataset to do 1000-way object classification. All convolutional filters are 3 × 3. (**B**) Modified architecture for feature-based attention tasks. To perform our feature-based attention tasks, the final layer that was implementing 1000-way softmax classification is replaced by binary classifiers (logistic regression), one for each category tested (two shown here, 20 total). These binary classifiers are trained on standard ImageNet images. (**C**) Test images for feature-based attention tasks. Merged images (left) contain two transparently overlaid ImageNet images of different categories. Array images (right) contain four ImageNet images on a 2 × 2 grid. Both are 224 × 224 pixels. These images are fed into the network and the binary classifiers are used to label the presence or absence of the given category. (**D**) Performance of binary classifiers. Box plots describe values over 20 different object categories (median marked in red, box indicates lower to upper quartile values and whiskers extend to full range, with the exception of outliers marked as dots). 'Standard' images are regular ImageNet images not used in the binary classifier training set.

DOI: https://doi.org/10.7554/eLife.38105.003

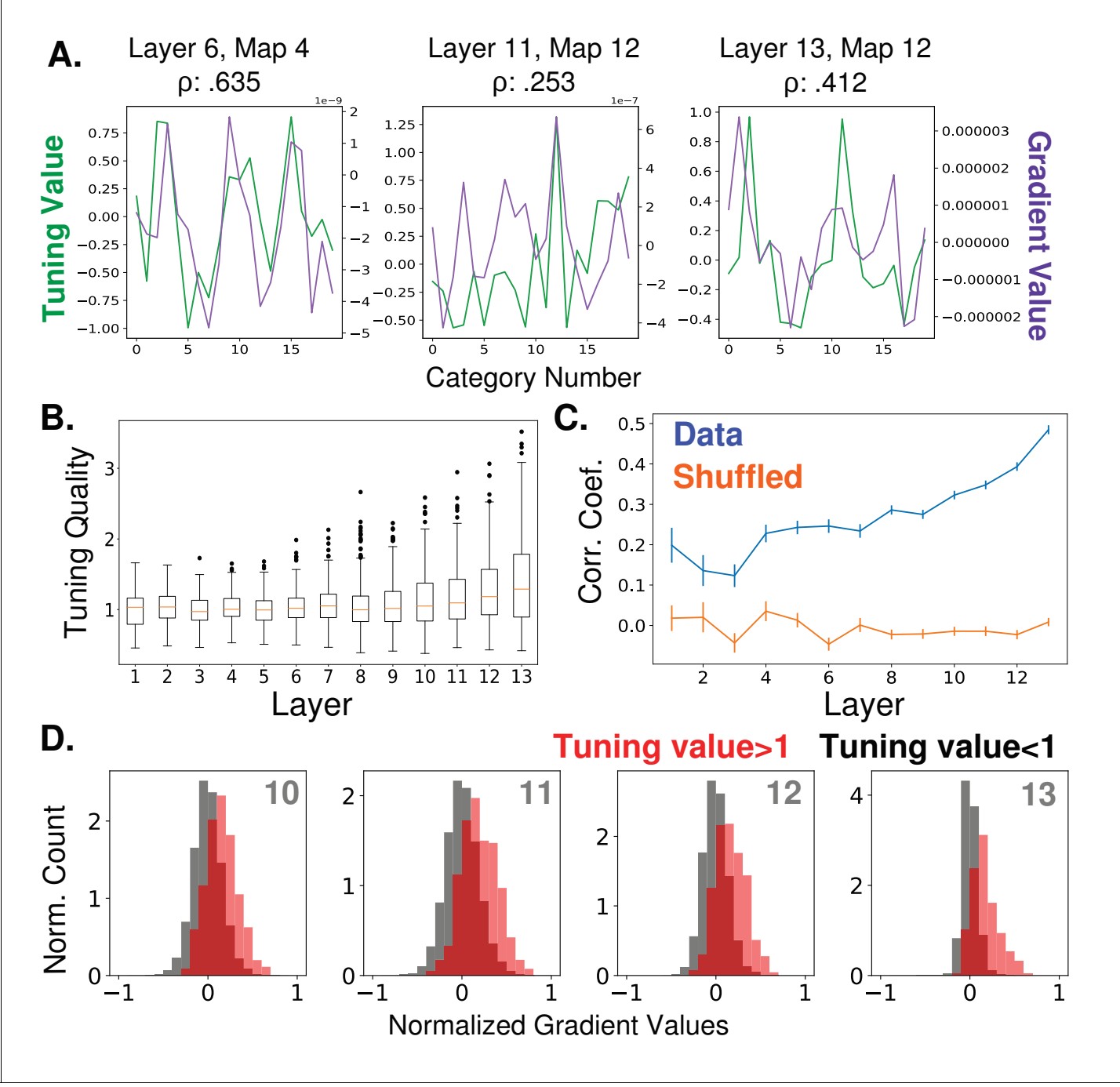

**Figure 2.** Relationship between feature map tuning and gradient values. (A) Example tuning values (green, left axis) and gradient values (purple, right axis) of three different feature maps from three different layers (identified in titles, layers as labelled in *Figure 1A*) over the 20 tested object categories. Tuning values indicate how the response to a category differs from the mean response; gradient values indicate how activity should change in order to classify input as from the category. Correlation coefficients between tuning curves and gradient values given in titles. All gradient and tuning values available in *Figure 2—source data 1* (B) Tuning quality across layers. Tuning quality is defined per feature map as the maximum absolute tuning value of that feature map. Box plots show distribution across feature maps for each layer. Average tuning quality for shuffled data: .372 ± .097 (this value does not vary significantly across layers) (C) Correlation coefficients between tuning curves and gradient value curves averaged over feature maps and plotted across layers (errorbars ± S.E.M., data values in blue and shuffled controls in orange). (D) Distributions of gradient values when tuning is strong. In red, histogram of gradient values associated with tuning values larger than one (i.e. for feature maps that strongly prefer the category), across all feature maps in layers 10, 11, 12, and 13. For comparison, histograms of gradient values associated with tuning values less than one are shown in black (counts are separately normalized for visibility, as the population in black is much larger than that in red).

*Figure 2 continued on next page*

*Figure 2 continued*

DOI: https://doi.org/10.7554/eLife.38105.004

The following source data is available for figure 2:

**Source data 1.** Object tuning curves and gradients.

DOI: https://doi.org/10.7554/eLife.38105.005

segregated according to tuning value. In red are gradient values that correspond to tuning values greater than one (for example, category 12 for the feature map in the middle pane of *Figure 2A*). As these distributions show, strong tuning values can be associated with weak or even negative gradient values. Negative gradient values indicate that increasing the activity of that feature map makes the network less likely to categorize the image as the given category. Therefore, even feature maps that strongly prefer a category (and are only a few layers from the classifier) still may not be involved in its classification, or even be inversely related to it. This is aligned with a recent neural network ablation study that shows category selectivity does not predict impact on classification (*Morcos et al., 2018*).

## Feature-based attention improves performance on challenging object classification tasks

To determine if manipulation according to tuning values can enhance performance, we created challenging visual images composed of multiple objects for the network to classify. These test images are of two types: merged (two object images transparently overlaid, such as in *Serences et al., 2004*) or array (four object images arranged on a grid) (see *Figure 1C* examples). The task for the network is to detect the presence of a given object category in these images. It does so using a series of binary classifiers trained on standard images of these objects, which replace the last layer of the network (*Figure 1B*). The performance of these classifiers on the test images indicates that this is a challenging task for the network (64.4% on merged images and 55.6% on array, *Figure 1D*. Chance is 50%), and thus a good opportunity to see the effects of attention.

We implement feature-based attention in this network by modulating the activity of units in each feature map according to how strongly the feature map prefers the attended object category (see Materials and methods, 'Tuning values' and 'How attention is applied'). A schematic of this is shown in *Figure 3A*. The slope of the activation function of units in a given feature map is scaled according to the tuning value of that feature map for the attended category (positive tuning values increase the slope while negative tuning values decrease it). Thus the impact of attention on activity is multiplicative and bi-directional.

The effects of attention are measured when attention is applied in this way at each layer individually (*Figure 3B*; solid lines) or all layers simultaneously (*Figure 3—figure supplement 1A*, red). For both image types (merged and array), attention enhances performance and there is a clear increase in performance enhancement as attention is applied at later layers in the network (numbering is as in *Figure 1A*). In particular, attention applied at the final convolutional layer performs best, leading to an 18.8% percentage point increase in binary classification on the merged images task and 22.8% increase on the array images task. Thus, FSGM-like effects can have large beneficial impacts on performance.

Attention applied at all layers simultaneously does not lead to better performance than attention applied at any individual layer (*Figure 3—figure supplement 1A*). We also performed a control experiment to ensure that nonspecific scaling of activity does not alone enhance performance (*Figure 3—figure supplement 1C*).

Some components of the FSGM are debated, for example whether attention impacts responses multiplicatively or additively (*Boynton, 2009*; *Baruni et al., 2015*; *Luck et al., 1997*; *McAdams and Maunsell, 1999* ), and whether the activity of cells that do not prefer the attended stimulus is actually suppressed (*Bridwell and Srinivasan, 2012*; *Navalpakkam and Itti, 2007*). Comparisons of different variants of the FSGM can be seen in *Figure 3—figure supplement 2*. In general, multiplicative and bidirectional effects work best.

We also measure performance when attention is applied using gradient values rather than tuning values (these gradient values are calculated to maximize performance on the binary classification

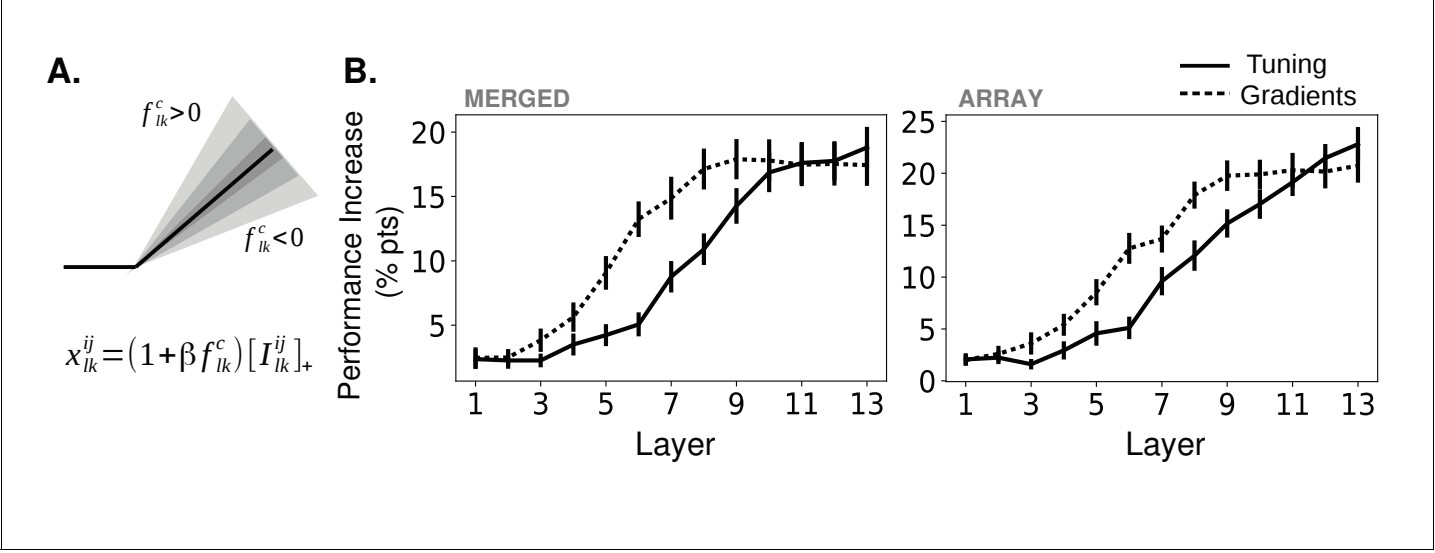

**Figure 3.** Effects of applying feature-based attention on object category tasks. (**A**) Schematic of how attention modulates the activity function. All units in a feature map are modulated the same way. The slope of the activation function is altered based on the tuning (or gradient) value, $f_{lk}^c$, of a given feature map (here, the $k^{th}$ feature map in the $l^{th}$ layer) for the attended category, $c$, along with an overall strength parameter $\beta$. $I_{lk}^{ij}$ Is the input to this unit from the previous layer. For more information, see Materials and methods, 'How attention is applied'. (**B**) Average increase in binary classification performance as a function of layer at which attention is applied (solid line represents using tuning values, dashed line using gradient values, errorbars ± S.E.M.). In all cases, best performing strength from the range tested is used for each instance. Performance shown separately for merged (left) and array (right) images. Gradients perform significantly ($p<.05$, $N = 20$) better than tuning at layers 5 – 8 (p=4.6e$^{-3}$, 2.6e$^{-5}$, 6.5e$^{-3}$, 4.4e$^{-3}$) for merged images and 5 – 9 (p=3.1e$^{-2}$, 2.3e$^{-4}$, 4.2e$^{-2}$, 6.1e$^{-3}$, 3.1e$^{-2}$) for array images. Raw performance values in *Figure 3—source data 1*.

DOI: https://doi.org/10.7554/eLife.38105.006

The following source data and figure supplements are available for figure 3:

**Source data 1.** Performance changes with attention.
DOI: https://doi.org/10.7554/eLife.38105.009

**Figure supplement 1.** Effect of applying attention to all layers or all feature maps uniformly.
DOI: https://doi.org/10.7554/eLife.38105.007

**Figure supplement 2.** Alternative forms of attention.
DOI: https://doi.org/10.7554/eLife.38105.008

task, rather than classify the image as a given category; therefore technically they differ from those shown in *Figure 2*, however in practice they are strongly correlated. See Materials and methods, 'Object category gradient calculations' and 'Gradient values' for details). Attention applied using gradient values shows the same layer-wise trend as when using tuning values. It also reaches the same performance enhancement peak when attention is applied at the final layers. The major difference, however, comes when attention is applied at middle layers of the network. Here, attention applied according to gradient values outperforms that of tuning values.

## Attention strength and the trade-off between increasing true and false positives

In the previous section, we examined the best possible effects of attention by choosing the strength for each layer and category that optimized performance. Here, we look at how performance changes as we vary the overall strength ($\beta$) of attention.

In *Figure 4A* we break the binary classification performance into true and false positive rates. Here, each colored line indicates a different category and increasing dot size represents increasing strength of attention. Ideally, true positives would increase without an equivalent increase (and possibly with a decrease) in false positive rates. If they increase in tandem, attention does not have a net beneficial effect. Looking at the effects of applying attention at different layers, we can see that attention at lower layers is less effective at moving the performance in this space and that movement is in somewhat random directions, although there is an average increase in performance with

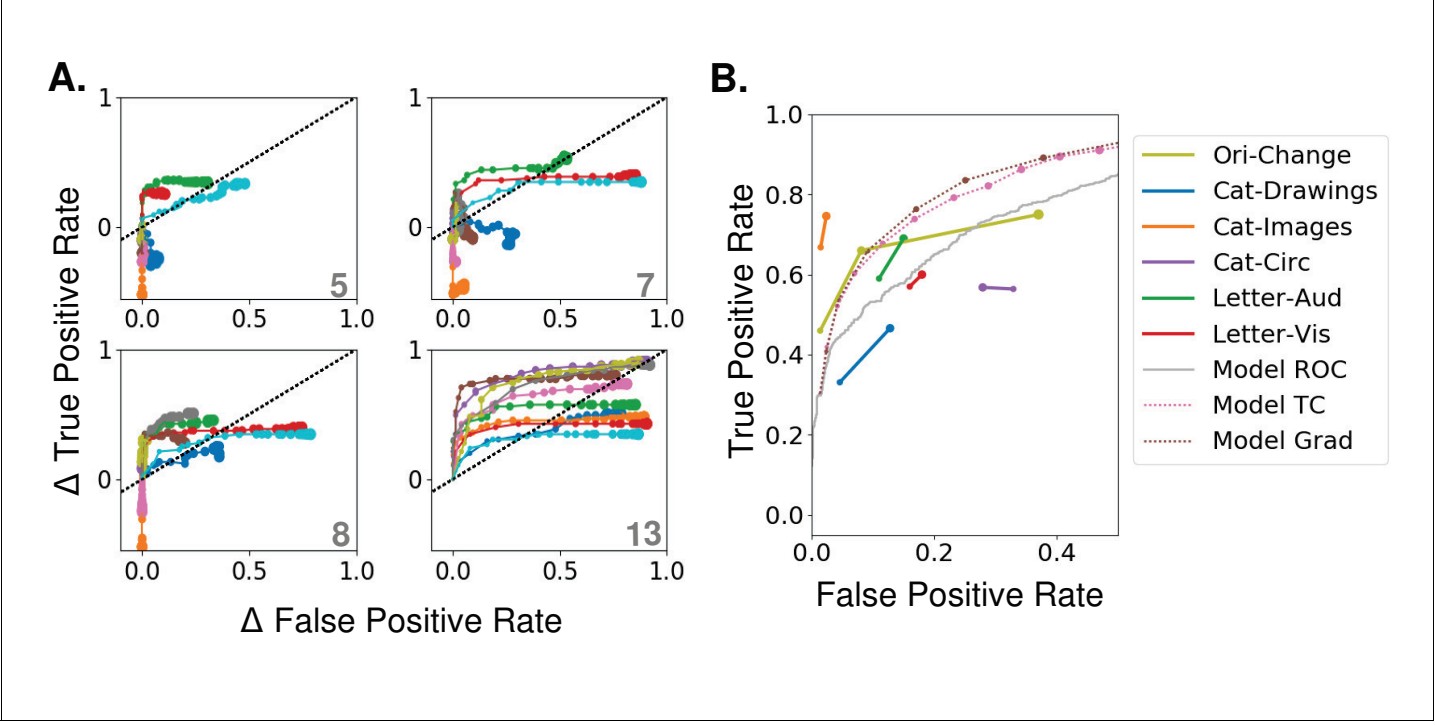

**Figure 4.** Effects of varying attention strength (**A**) Effect of increasing attention strength (*β*) in true and false positive rate space for attention applied at each of four layers (layer indicated in bottom right of each panel, attention applied using tuning values). Each line represents performance for an individual category (only 10 categories shown for visibility), with each increase in dot size representing a .15 increase in *β*. Baseline (no attention) values are subtracted for each category such that all start at (0,0). The black dotted line represents equal changes in true and false positive rates. (**B**) Comparisons from experimental data. The true and false positive rates from six experiments in four previously published studies are shown for conditions of increasing attentional strength (solid lines). Cat-Drawings = (***Lupyan and Ward, 2013***), Exp. 1; Cat-Images=(***Lupyan and Ward, 2013***), Exp. 2; Objects=(***Koivisto and Kahila, 2017***), Letter-Aud.=(***Lupyan and Spivey, 2010***), Exp. 1; Letter-Vis.=(***Lupyan and Spivey, 2010***), Exp. 2. Ori-Change=(***Mayo and Maunsell, 2016***). See Materials and methods, 'Experimental data' for details of experiments. Dotted lines show model results for merged images, averaged over all 20 categories, when attention is applied using either tuning (TC) or gradient (Grad) values at layer 13. Model results are shown for attention applied with increasing strengths (starting at 0, with each increasing dot size representing a .15 increase in *β*). Receiver operating curve (ROC) for the model using merged images, which corresponds to the effect of changing the threshold in the final, readout layer, is shown in gray. Raw performance values in ***Figure 3—source data 1***.

DOI: https://doi.org/10.7554/eLife.38105.010

The following figure supplement is available for figure 4:

**Figure supplement 1.** Negatively applying attention and best-performing strengths.

DOI: https://doi.org/10.7554/eLife.38105.011

moderate attentional strength. With attention applied at later layers, true positive rates are more likely to increase for moderate attentional strengths, while substantial false positive rate increases occur only with higher strengths. Thus, when attention is applied with modest strength at layer 13, most categories see a substantial increase in true positives with only modest increases in false positives. As strength continues to increase however, false positives increase substantially and eventually lead to a net decrease in overall classifier performance (representing as crossing the dotted line in *Figure 4A*).

Applying attention according to negated tuning values leads to a decrease in true and false positive values with increasing attention strength, which decreases overall performance (*Figure 4—figure supplement 1A*). This verifies that the effects of attention are not from non-specific changes in activity.

Experimentally, when switching from no or neutral attention, neurons in MT showed an average increase in activity of 7% when attending their preferred motion direction (and similar decrease when attending the non-preferred) (*Martinez-Trujillo and Treue, 2004*). In our model, when *β* = .75 (roughly the value at which performance peaks at later layers; *Figure 4—figure supplement 1B*),

given the magnitude of the tuning values (average magnitude: .38), attention scales activity by an average of 28.5%. This value refers to how much activity is modulated in comparison to the $\beta = 0$ condition, which is probably more comparable to passive or anesthetized viewing, as task engagement has been shown to scale neural responses generally (*Page and Duffy, 2008*). This complicates the relationship between modulation strength in our model and the values reported in the data.

To allow for a more direct comparison, in *Figure 4B*, we collected the true and false positive rates obtained experimentally during different object detection tasks (explained in Materials and methods, 'Experimental data'), and plotted them in comparison to the model results when attention is applied at layer 13 using tuning values (pink line) or gradient value (brown line). Five experiments (second through sixth studies) are human studies. In all of these, uncued trials are those in which no information about the upcoming visual stimulus is given, and therefore attention strength is assumed to be low. In cued trials, the to-be-detected category is cued before the presentation of a challenging visual stimulus, allowing attention to be applied to that object or category.

The majority of these experiments show a concurrent increase in both true and false positive rates as attention strength is increased. The rates in the uncued conditions (smaller dots) are generally higher than the rates produced by the $\beta = 0$ condition in our model, consistent with neutrally cued conditions corresponding to $\beta > 0$. We find (see Materials and methods, 'Experimental data'), that the average corresponding $\beta$ value for the neutral conditions is .37 and for the attended conditions .51. Because attention scales activity by $1 + \beta f_c^{lk}$ (where $f_c^{lk}$ is the tuning value), these changes correspond to a $\approx 5\%$ change in activity.

The first dataset included in the plot (Ori-Change; yellow line in *Figure 4B*) comes from a macaque change detection study (see Materials and methods, 'Experimental data' for details). Because the attention cue was only 80% valid, attention strength could be of three levels: low (for the uncued stimuli on cued trials), medium (for both stimuli on neutrally-cued trials), or high (for the cued stimuli on cued trials). Like the other studies, this study shows a concurrent increase in both true positive (correct change detection) and false positive (premature response) rates with increasing attention strength. For the model to achieve the performance changes observed between low and medium attention a roughly 12% activity change is needed, but average V4 firing rates recorded during this task show an increase of only 3.6%. This discrepancy may suggest that changes in correlations (*Cohen and Maunsell, 2009*) or firing rate changes in areas aside from V4 also make important contributions to observed performance changes.

Thus, according to our model, the size of experimentally observed performance changes is broadly consistent with the size of experimentally observed neural changes. While other factors are likely also relevant for performance changes, this rough alignment between the magnitude of firing rate changes and magnitude of performance changes supports the idea that the former could be a major causal factor for the latter. In addition, the fact that the model can capture this relationship provides further support for its usefulness as a model of the biology.

Finally, we show the change in true and false positive rates when the threshold of the final layer binary classifier is varied (a 'receiver operating characteristic' analysis, *Figure 4B*, gray line; no attention was applied during this analysis). Comparing this to the pink line, it is clear that varying the strength of attention applied at the final convolutional layer has more favorable performance effects than altering the classifier threshold (which corresponds to an additive effect of attention at the classifier layer). This points to the limitations that could come from attention targeting only downstream readout areas.

Overall, the model roughly matches experiments in the amount of neural modulation needed to create the observed changes in true and false positive rates. However, it is clear that the details of the experimental setup are relevant, and changes aside from firing rate and/or outside the ventral stream also likely play a role (*Navalpakkam and Itti, 2007*).

## Feature-based attention enhances performance on orientation detection task

Some of the results presented above, particularly those related to the layer at which attention is applied, may be influenced by the fact that we are using an object categorization task. To see if results are comparable using the simpler stimuli frequently used in macaque studies, we created an orientation detection task (*Figure 5A*). Here, binary classifiers trained on full-field oriented gratings

are tested using images that contain two gratings of different orientation and color. The performance of these binary classifiers without attention is above chance (distribution across orientations shown in inset of *Figure 5A*). The performance of the binary classifier associated with vertical orientation (0 degrees) was abnormally high (92% correct without attention, other orientations average 60.25%. This likely reflects the over-representation of vertical lines in the training images) and this orientation was excluded from further performance analysis.

Attention is applied according to orientation tuning values of the feature maps (tuning quality by layer is shown in *Figure 5B*) and tested across layers. We find (*Figure 5D*, solid line and *Figure 3— figure supplement 1B*, red) that the trend in this task is similar to that of the object task: applying attention at later layers leads to larger performance increases (14.4% percentage point increase at layer 10). This is despite the fact that orientation tuning quality peaks in the middle layers.

We also calculate the gradient values for this orientation detection task. While overall the correlations between gradient values and tuning values are lower (and even negative for early layers), the average correlation still increases with layer (*Figure 5C*), as with the category detection task. Importantly, while this trend in correlation exists in both detection tasks tested here, it is not a universal feature of the network or an artifact of how these values are calculated. Indeed, an opposite pattern in the correlation between orientation tuning and gradient values is shown when using attention to orientation to classify the color of a stimulus with the attended orientation (see 'Recordings show how feature similarity gain effects propagate', and Materials and methods, 'Oriented grating attention tasks' and 'Gradient values').

The results of applying attention according to gradient values is shown in *Figure 5D* (dashed line). Here again, using gradient value creates similar trends as using tuning values, with gradient values performing better in the middle layers.

## Feature-based attention primarily influences criteria and spatial attention primarily influences sensitivity

Signal detection theory is frequently used to characterize the effects of attention on performance (*Verghese, 2001*). Here, we use a joint feature-spatial attention task to explore effects of attention in the model. The task uses the same two-grating stimuli described above. The same binary orientation classifiers are used and the task of the model is to determine if a given orientation is present in a given quadrant of the image. Performance is then measured when attention is applied to an orientation, a quadrant, or both an orientation and a quadrant (effects are combined additively, for more, see Materials and methods, 'How attention is applied'). Two key signal detection measurements are computed: criteria and sensitivity. Criteria is a measure of the threshold that's used to mark an input as positive, with a higher criteria leading to fewer positives; sensitivity is a measure of the separation between the two populations (positives and negatives), with higher sensitivity indicating a greater separation.

*Figure 5E* shows that both spatial and feature-based attention influence sensitivity and criteria. However, feature-based attention decreases criteria more than spatial attention does. Intuitively, feature-based attention shifts the representations of all stimuli in the direction of the attended category, implicitly lowering the detection threshold. Starting from a high threshold, this can lead to the observed behavioural pattern wherein true positives increase before false positives do. Sensitivity increases more for spatial attention alone than for feature-based attention alone, indicating that spatial attention amplifies differences in the representation of whichever features are present. These general trends hold regardless of the layer at which attention is applied and whether feature-based attention is applied using tuning curves or gradients. Changes in true and false positive rates for this task can be seen explicitly in *Figure 5—figure supplement 1*.

In line with our results, spatial attention was found experimentally to increase sensitivity and (less reliably) decrease criteria (*Hawkins et al., 1990*; *Downing, 1988*). Furthermore, feature-based attention is known to decrease criteria, with lesser effects on sensitivity (*Rahnev et al., 2011*; *Bang and Rahnev, 2017*; though see *Stein and Peelen, 2015*). A study that looked explicitly at the different effects of spatial and category-based attention (*Stein and Peelen, 2017*) found that spatial attention increases sensitivity more than category-based attention (most visible in their Experiment 3c, which uses natural images), and the effects of the two are additive.

Attention and priming are known to impact neural activity beyond pure sensory areas (*Krauzlis et al., 2013*; *Crapse et al., 2018*). This idea is borne out by a study that aimed to isolate

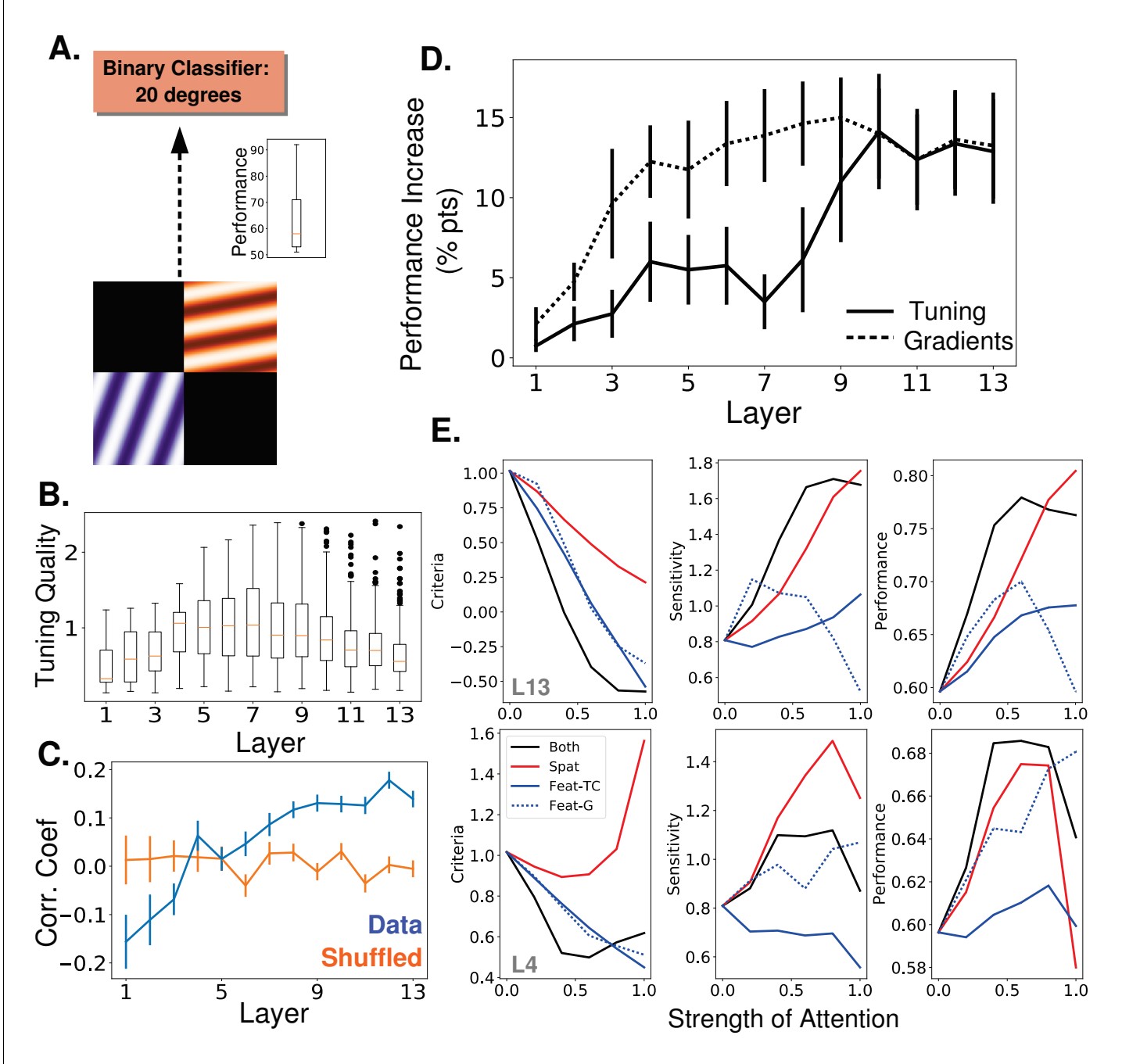

**Figure 5.** Attention task and results using oriented gratings. (A) Orientation detection task. Like with the object category detection tasks, separate binary classifiers trained to detect each of 9 different orientations replaced the final layer of the network. Test images included two oriented gratings of different color and orientation located at 2 of 4 quadrants. Inset shows performance over nine orientations without attention (B) Orientation tuning quality as a function of layer. (C) Average correlation coefficient between orientation tuning curves and gradient curves across layers (blue). Shuffled correlation values in orange. Errorbars are ± S.E.M. (D) Comparison of performance on orientation detection task when attention is determined by tuning values (solid line) or gradient values (dashed line) and applied at different layers. As in *Figure 3B*, best performing strength is used in all cases. Errorbars are ±S.E.M. Gradients perform significantly (p=1.9e -2) better than tuning at layer 7. Raw performance values available in *Figure 5—source data 1*. (E) Change in signal detection values and performance (perent correct) when attention is applied in different ways—spatial (red), feature according to tuning (solid blue), feature according to gradients (dashed blue), and both spatial and feature (according to tuning, black)—for the task of detecting a given orientation in a given quadrant. Top row is when attention is applied at layer 13 and bottom when applied at layer 4. Raw performance values available in *Figure 5—source data 2*.

DOI: https://doi.org/10.7554/eLife.38105.012

*Figure 5 continued on next page*

*Figure 5 continued*

The following source data and figure supplement are available for figure 5:

**Source data 1.** Performance on orientation detection task.

DOI: https://doi.org/10.7554/eLife.38105.014

**Source data 2.** Performance on spatial and feature-based attention task.

DOI: https://doi.org/10.7554/eLife.38105.015

**Figure supplement 1.** True and false positive changes with spatial and feature-based attention.

DOI: https://doi.org/10.7554/eLife.38105.013

the neural changes associated with sensitivity and criteria changes (*Luo and Maunsell, 2015*) In this study, the authors designed behavioural tasks that encouraged changes in behavioural sensitivity or criteria exclusively: high sensitivity was encouraged by associating a given stimulus location with higher overall reward, while high criteria was encouraged by rewarding correct rejects more than hits (and vice versa for low sensitivity/criteria). Differences in V4 neural activity were observed between trials using high versus low sensitivity stimuli. No differences were observed between trials using high versus low criteria stimuli. This indicates that areas outside of the ventral stream (or at least outside V4) are capable of impacting criteria (*Sridharan et al., 2017*). Importantly, it does not mean that changes in V4 don't impact criteria, but merely that those changes can be countered by the impact of changes in other areas. Indeed, to create sessions wherein sensitivity was varied without any change in criteria, the authors had to increase the relative correct reject reward (i.e., increase the criteria) at locations of high absolute reward, which may have been needed to counter a decrease in criteria induced by attention-related changes in V4 (similarly, they had to decrease the correct reject reward at low reward locations). Our model demonstrates clearly how such effects from sensory areas alone can impact detection performance, which, in turn highlights the role downstream areas may play in determining the final behavioural outcome.

## Recordings show how feature similarity gain effects propagate

To explore how attention applied at one location in the network impacts activity later on, we apply attention at various layers and 'record' activity at others (*Figure 6A*, in response to full field oriented gratings). In particular, we record activity of feature maps at all layers while applying attention at layers 2, 6, 8, 10, or 12 individually.

To understand the activity changes occurring at each layer, we use an analysis from (*Martinez-Trujillo and Treue, 2004*) that was designed to test for FSGM-like effects and is explained in *Figure 6B*. Here, the activity of a feature map in response to a given orientation when attention is applied is divided by the activity in response to the same orientation without attention. These ratios are organized according to the feature map's orientation preference (most to least) and a line is fit to them. According to the FSGM of attention, this ratio should be greater than one for more preferred orientations and less than one for less preferred, creating a line with an intercept greater than one and negative slope.

In *Figure 6C*, we plot the median value of the slopes and intercepts across all feature maps at a layer, when attention is applied at different layers (indicated by color). When attention is applied directly at a layer according to its tuning values (left), FSGM effects are seen by default (intercept values are plotted in terms of how they differ from one; comparable average values from (*Martinez-Trujillo and Treue, 2004*) are intercept: .06 and slope: 0.0166, but note we are using $\beta = 0$ for the no-attention condition in the model which, as mentioned earlier, is not necessarily the best analogue for no-attention conditions experimentally. Therefore we use these measures to show qualitative effects). As these activity changes propagate through the network, however, the FSGM effects wear off, suggesting that activating units tuned for a stimulus at one layer does not necessarily activate cells tuned for that stimulus at the next. This misalignment between tuning at one layer and the next explains why attention applied at all layers simultaneously isn't more effective (*Figure 3—figure supplement 1*). In fact, applying attention to a category at one layer can actually have effects that counteract attention at a later layer (see *Figure 6—figure supplement 1*).

In *Figure 6C* (right), we show the same analysis, but while applying attention according to gradient values. The effects at the layer at which attention is applied do not look strongly like FSGM,

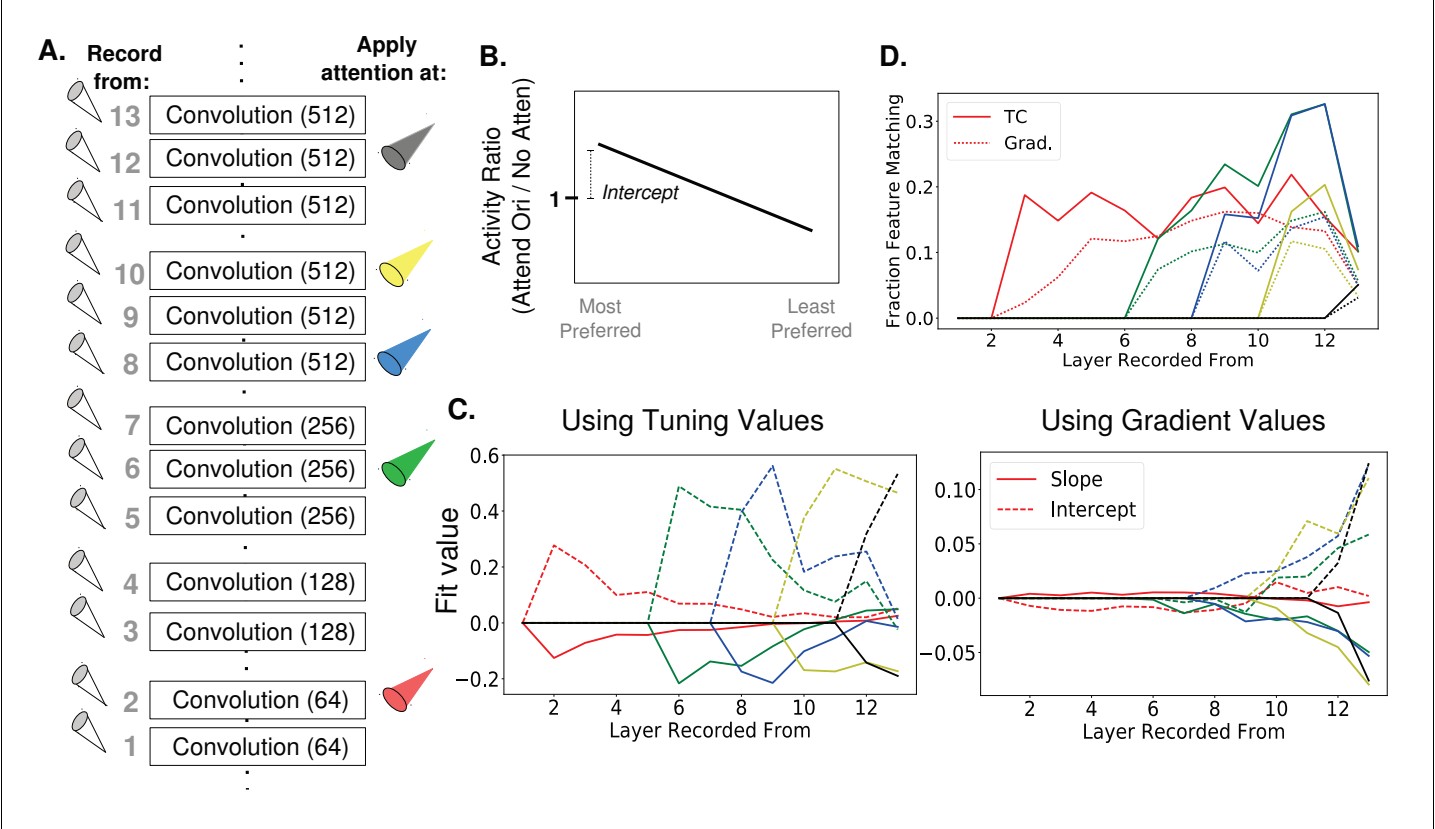

**Figure 6.** How attention-induced activity changes propagate through the network. (**A**) Recording setup. The spatially averaged activity of feature maps at each layer was recorded (left) while attention was applied at layers 2, 6, 8, 10, or 12 individually. Activity was in response to a full field oriented grating. (**B**) Schematic of metric used to test for the feature similarity gain model. Activity when a given orientation is present and attended is divided by the activity when no attention is applied, giving a set of activity ratios. Ordering these ratios from most to least preferred orientation and fitting a line to them gives the slope and intercept values plotted in (**C**). Intercept values are plotted in terms of how they differ from 1, so positive values are an intercept greater than 1. (FSGM predicts negative slope and positive intercept). (**C**) The median slope (solid line) and intercept (dashed line) values as described in (**B**) plotted for each layer when attention is applied to the layer indicated by the line color as labelled in (**A**). On the left, attention applied according to tuning values and on the right, attention applied according to gradient values. Raw slope and intercept values when using tuning curves available in *Figure 6—source data 1* and for gradients in *Figure 6—source data 2*. (**D**) Fraction of feature maps displaying feature matching behaviour at each layer when attention is applied at the layer indicated by line color. Shown for attention applied according to tuning (solid lines) and gradient values (dashed line).

DOI: https://doi.org/10.7554/eLife.38105.016

The following source data and figure supplements are available for figure 6:

**Source data 1.** Intercepts and slopes from gradient-applied attention.
DOI: https://doi.org/10.7554/eLife.38105.019
**Source data 2.** Intercepts and slopes from tuning curve-applied attention.
DOI: https://doi.org/10.7554/eLife.38105.020
**Figure supplement 1.** Feature-based attention at one layer often suppresses activity of the attended features at later layers.
DOI: https://doi.org/10.7554/eLife.38105.017
**Figure supplement 2.** Correlating activity changes with performance changes.
DOI: https://doi.org/10.7554/eLife.38105.018

however FSGM properties evolve as the activity changes propagate through the network, leading to clear FSGM-like effects at the final layer. Finding FSGM-like behaviour in neural data could thus be a result of FSGM effects at that area or non-FSGM effects at an earlier area (here, attention applied according to gradients which, especially at earlier layers, are not aligned with tuning).

An alternative model of the neural effects of attention—the feature matching (FM) model—suggests that the effect of attention is to amplify the activity of a neuron whenever the stimulus in its receptive field matches the attended stimulus. In *Figure 6D*, we calculate the fraction of feature

maps at a given layer that show feature matching behaviour (defined as having activity ratios greater than one when the stimulus orientation matches the attended orientation for both preferred and anti-preferred orientations). As early as one layer post-attention, some feature maps start showing feature matching behaviour. The fact that the attention literature contains conflicting findings regarding the feature similarity gain model versus the feature matching model (*Motter, 1994*; *Ruff and Born, 2015*) may result from this finding that FSGM effects can turn into FM effects as they propagate through the network. In particular, this mechanism can explain the observations that feature matching behaviour is observed more in FEF than V4 (*Zhou and Desimone, 2011*) and that match information is more easily read out from perirhinal cortex than IT (*Pagan et al., 2013*).

We also investigated the extent to which measures of attention's neural effects correlate with changes in performance (see Materials and methods, 'Correlating activity changes with performance'). For this we developed a new, experimentally-feasible way of calculating attention's effects on neural activity that is inspired by the gradient-based approach to attention (that is, it focuses on classification rather than tuning). We show (*Figure 6—figure supplement 2*) that this new measure better correlates with performance changes than the FSGM measure of activity changes, particularly at earlier layers.

There is a simple experiment that would distinguish whether factors beyond tuning, such as gradients, play a role in guiding attention. It requires using two tasks with very different objectives (which should produce different gradients) but with the same attentional cue. An example is described in *Figure 7*. Here, the two tasks used would be an orientation-based color classification task (two gratings each with their own color and orientation are simultaneously shown, and the task is to report the color of the grating with the attended orientation) and an orientation detection task (report if the attended orientation is present or absent in the image). In both cases, attention is cued according to orientation. But gradient-based attention will produce different neural modulations for the two tasks, while the FSGM predicts identical modulations (*Figure 7C*). Thus, an experiment that recorded from the same neurons during both tasks could distinguish between tuning-based and gradient-based attention.

## Discussion

In this work, we utilized a deep convolutional neural network (CNN) as a model of the visual system to probe the relationship between modulation of neural activity, as in attention, and performance. Specifically, we formally define the feature similarity gain model (FSGM) of attention (the basic tenets of which have been described in several experimental studies) as a multiplicative modulation of neuronal activity proportional to the neuron's mean-subtracted feature tuning. This formalization allows us to investigate the FSGM's ability to enhance a CNN's performance on challenging visual tasks. We found that, across a variety of tasks, neural activity changes matching the type and magnitude of those observed experimentally can indeed lead to performance changes of the kind and magnitude observed experimentally.

We used the full observability of the model to investigate the relationship between tuning and function. We compared attention applied according to feature tuning (the FSGM) with attention designed to optimally modulate activity to improve performance (as determined by the gradient of performance with respect to the neural activity). Attention applied according to tuning does not successfully propagate from lower or middle to higher layers; that is, enhancing the activity of neurons that most prefer a given category at lower layers need not selectively enhance the activity of neurons preferring that category at higher layers. As a result, attention applied according to the FSGM performs poorly when applied at early to middle layers, while attention applied according to gradients at these layers performs better.

Attention is most effective applied at later layers (e.g., layers 9–13), where tuning and gradient values are better correlated. According to (*Güçlü and van Gerven, 2015*), these layers correspond most to areas V4 and LO. Such areas are known and studied for reliably showing attentional effects, whereas earlier areas such as V1 are generally not (*Luck et al., 1997*; *Abdelhack and Kamitani, 2018*). In a study involving detection of objects in natural scenes, the strength of category-specific preparatory activity in object selective cortex was correlated with performance, whereas such preparatory activity in V1 was anti-correlated with performance (*Peelen and Kastner, 2011*). This is in line

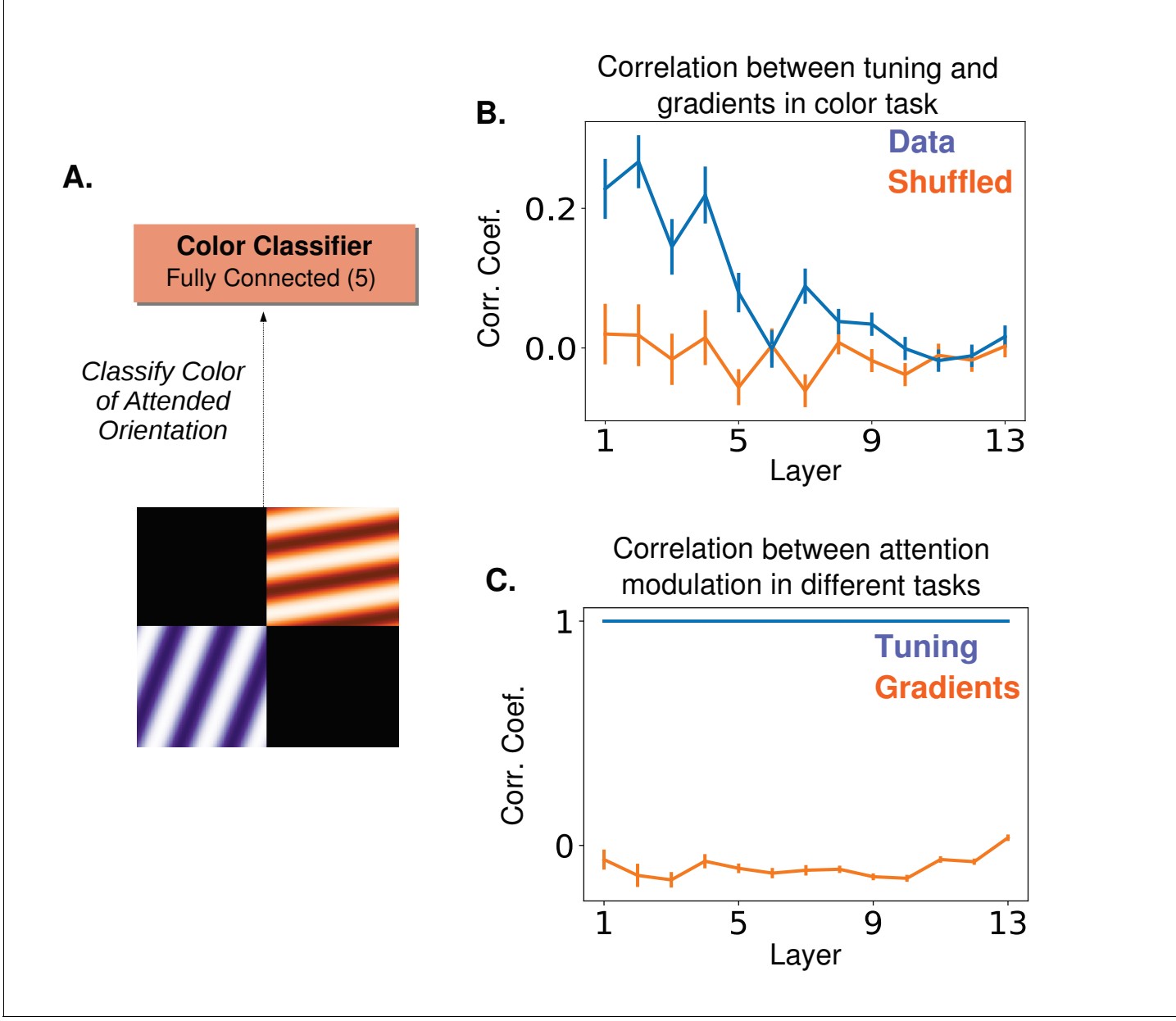

**Figure 7.** A proposed experiment to distinguish between tuning-based and gradient-based attention  (A) 'Cross-featural' attention task. Here, the final layer of the network is replaced with a color classifier and the task is to classify the color of the attended orientation in a two-orientation stimulus. Importantly, in both this and the orientation detection task (*Figure 5A*), a subject performing the task would be cued to attend to an orientation. (B) The correlation coefficient between the gradient values calculated for this task and orientation tuning values (as in *Figure 5C*). Correlation peaks at lower layers for this task. (C) Correlation between tuning values for the two tasks (blue) and between gradient values for the two tasks (orange). If attention does target cells based on tuning, the modulation would be the same in both the color classification task and the orientation detection task. If a gradient-based targeting is used, no (or even a slight anti-) correlation is expected. Tuning and gradient values available in *Figure 7—source data 1*.

DOI: https://doi.org/10.7554/eLife.38105.021
The following source data is available for figure 7:

**Source data 1.** Orientation tuning curves and gradients.
DOI: https://doi.org/10.7554/eLife.38105.022

with our finding that feature-based attention effects at earlier areas can counter the beneficial effects of that attention at later areas (**Figure 6—figure supplement 1**).

Our work raises the question: is attention applied simply according to tuning or is it targeted to best optimize function on a given task? We suggested a simple experiment (**Figure 7**) that would reveal whether non-tuning factors, such as gradients, guide attentional modulation. In (**Chelazzi et al., 1998**) the correlation coefficient between an index of tuning and an index of attentional modulation was .52 for a population of V4 neurons, suggesting factors other than selectivity influence attention. Furthermore, many attention studies, including that one, use only preferred and anti-preferred stimuli and therefore don't include a thorough investigation of the relationship between tuning and attentional modulation. (**Martinez-Trujillo and Treue, 2004**) uses multiple stimuli to provide support for the FSGM, however the interpretation is limited by the fact that they only report population averages. (**Ruff and Born, 2015**) investigated the relationship between tuning strength and the strength of attentional modulation on a cell-by-cell basis. While they did find a correlation (particularly for binocular disparity tuning), it was relatively weak, which leaves room for the possibility that tuning is not the primary factor that determines attentional modulation. Local connectivity is also likely to play a role, as a correlation between normalization and attentional modulation has been shown (**Ni et al., 2012**).

A major challenge for understanding the biological implementation of selective attention is determining how such a precise attentional signal is carried by feedback connections. We believe that it is plausible that the visual system can learn the connections needed to carry out gradient-based attention. For example, if a high-level neuron related to the classification of an image sends a feedback connection to lower areas, an anti-Hebbian post-pre spike timing-dependent learning rule would strengthen the connection from the high level neuron to the low level one, if the lower level one causes the firing of the higher. In this way, neurons in later areas can learn to target the cells in earlier areas that caused them to fire. In contrast, it is actually more difficult to imagine how higher areas could learn the connections needed to target neurons according to their tuning, as in the FSGM. The machine learning literature on attention and learning may inspire other useful hypotheses on underlying brain mechanisms (**Xu et al., 2015**; **Lillicrap et al., 2016**).

The concept of attention has been introduced in these models previously in the machine learning literature (**Mnih et al., 2014**). Generally, this kind of attention relates to what would be called overt spatial attention in the neuroscience literature. That is, the attention mechanism serially selects areas of the input image for further processing, rather than modulating the activity of neurons representing those areas (as in our model of spatial attention). Other work has been done using attention to selectively process image features (**Stollenga et al., 2014**) and it would be interesting to compare the workings of that model to the feature-based attention used in our study.

While CNNs have representations that are similar to the ventral stream, they lack many biological details including recurrent connections, dynamics, cell types, and noisy responses. Preliminary work has shown that these elements can be incorporated into a CNN structure, and attention can enhance performance in this more biologically-realistic architecture (**Lindsay et al., 2017**). Furthermore, while the current work does not include neural noise independent of the stimulus, the fact that a given image is presented in many contexts (different merged images or different array images) can be thought of as a form of highly structured noise that does produce variable responses to the same image.

Another biological detail that this model lacks is 'skip connections,' where one layer feeds into both the layer directly after it and deeper layers after that (**He et al., 2016**; **Huang et al., 2017**) as in connections from V2 to V4 or V4 to parietal areas (**Ungerleider et al., 2008**). Our results regarding propagation of changes through the network suggest that synaptic distance from the classifier is a relevant feature—one that is less straight forward to determine in a network with skip connections.

Because experimenters can easily control the image, defining a cell's function in terms of how it responds to stimuli makes practical sense. However, it may be that thinking about visual areas in terms of their synaptic distance from decision-making areas such as prefrontal cortex (**Heekeren et al., 2004**) can be more useful for the study of attention than thinking in terms of their distance from the retina. Thus far, coarse stimulation protocols have found a relationship between the tuning of neural populations and their impact on perception (**Moeller et al., 2017**; **DeAngelis et al., 1998**; **Salzman et al., 1990**). However, studies of the relationship between tuning and choice probabilities suggest that a neuron's preferred stimulus is not always an indication of its

causal role in classification (*Zaidel et al., 2017*; *Purushothaman and Bradley, 2005*), though see (*Katz et al., 2016*). Targeted stimulation protocols and a more fine-grained ability to determine both upstream drivers of, and downstream responses driven by, stimulated neurons will be needed to better address these issues.

## Materials and methods

### Key resources
The weights for the model ('VGG-16') came from *Frossard (2017)* (RRID SCR_016494).

### Network model
This work uses a deep convolutional neural network (CNN) as a model of the ventral visual stream. Convolutional neural networks are feed forward artificial neural networks that consist of a few basic operations repeated in sequence, key among them being the convolution. The specific CNN architecture used in the study comes from *Simonyan and Zisserman, 2014* (VGG-16D) and is shown in *Figure 1A* (a previous variant of this work used a smaller network (*Lindsay, 2015*). For this study, all the layers of the CNN except the final classifier layer were pre-trained using back propagation on the ImageNet classification task, which involves doing 1000-way object categorization (weights provided by *Frossard, 2017*). The training of the top layer is described in subsequent sections. Here we describe the basic workings of the CNN model we use, with details available in *Simonyan and Zisserman, 2014*.

The activity values of the units in each convolutional layer are the result of applying a 2-D spatial convolution to the layer below, followed by positive rectification (rectified linear 'ReLu' nonlinearity):

$$x_{ij}^{lk} = \left[ \left( W^{lk} \star X^{l-1} \right)_{ij} \right]_+ \tag{1}$$

where $\star$ indicates convolution, and $[x]_+ = x$ if $x > 0$, 0 otherwise. $W^{lk}$ is the $k^{th}$ convolutional filter at the $l^{th}$ layer. The application of each filter results in a 2-D feature map (the number of filters used varies across layers and is given in parenthesis in *Figure 1A*). $x_{ij}^{lk}$ is the activity of the unit at the $i,j^{th}$ spatial location in the $k^{th}$ feature map at the $l^{th}$ layer. $X^{l-1}$ is thus the activity of all units at the layer below the $l^{th}$ layer. The input to the network is a 224 by 224 pixel RGB image, and thus the first convolution is applied to these pixel values. Convolutional filters are $3 \times 3$. For the purposes of this study the convolutional layers are most relevant, and will be referred to according to their numbering in *Figure 1A* (numbers in parentheses indicate number of feature maps per layer).

Max pooling layers reduce the size of the feature maps by taking the maximum activity value of units in a given feature map in non-overlapping $2 \times 2$ windows. Through this, the size of the feature maps decreases after each max pooling (layers 1 and 2: $224 \times 224$; 3 and 4: $112 \times 112$; 5, 6, and 7: $56 \times 56$. 8, 9, and 10: $28 \times 28$; 11, 12, and 13: $14 \times 14$).

The final two layers before the classifier are each fully-connected to the layer below them, with the number of units per layer given in parenthesis in *Figure 1A*. Therefore, connections exist from all units from all feature maps in the last convolutional layer (layer 13) to all 4096 units of the next layer, and so on. The top readout layer of the network in (*Simonyan and Zisserman, 2014*) contained 1000 units upon which a softmax classifier was used to output a ranked list of category labels for a given image. Looking at the top-5 error rate (wherein an image is correctly labelled if the true category appears in the top five categories given by the network), this network achieved 92.7% accuracy. With the exception of the gradient calculations described below, we did not use this 1000-way classifier, but rather replaced it with a series of binary classifiers.

### Object category attention tasks
The tasks we use to probe the effects of feature-based attention in this network involve determining if a given object category is present in an image or not, similar to tasks used in (*Stein and Peelen, 2017*; *Peelen et al., 2009*; *Koivisto and Kahila, 2017*). To have the network perform this specific task, we replaced the final layer in the network with a series of binary classifiers, one for each category tested (*Figure 1B*). We tested a total of 20 categories: paintbrush, wall clock, seashore,

paddlewheel, padlock, garden spider, long-horned beetle, cabbage butterfly, toaster, greenhouse, bakery, stone wall, artichoke, modem, football helmet, stage, mortar, consomme, dough, bathtub. Binary classifiers were trained using ImageNet images taken from the 2014 validation set (and were therefore not used in the training of the original model). A total of 35 unique true positive images were used for training for each category, and each training batch was balanced with 35 true negative images taken from the remaining 19 categories. The results shown here come from using logistic regression as the binary classifier, though trends in performance are similar if support vector machines are used.

Once these binary classifiers are trained, they are then used to classify more challenging test images. Experimental results suggest that classifiers trained on unattended and isolated object images are appropriate for reading out attended objects in cluttered images (*Zhang et al., 2011*). These test images are composed of multiple individual images (drawn from the 20 categories) and are of two types: 'merged' and 'array'. Merged images are generated by transparently overlaying two images, each from a different category (specifically, pixel values from each are divided by two and then summed). Array images are composed of four separate images (all from different categories) that are scaled down to 112 by 112 pixels and placed on a two by two grid. The images that comprise these test images also come from the 2014 validation set, but are separate from those used to train the binary classifiers. See examples of each in *Figure 1C*. Test image sets are balanced (50% do contain the given category and 50% do not, 150 total test images per category). Both true positive and true negative rates are recorded and overall performance is the average of these rates.

## Object category gradient calculations

When neural networks are trained via back propagation, gradients are calculated that indicate how a given weight in the network impacts the final classification. We use this same method to determine how a given unit's activity impacts the final classification. Specifically, we input a 'merged' image (wherein one of the images belongs to the category of interest) to the network. We then use gradient calculations to determine the changes in activity that would move the 1000-way classifier toward classifying that image as belonging to the category of interest (i.e. rank that category highest). We average these activity changes over images and over all units in a feature map. This gives a single value per feature map:

$$g_c^{lk} = -\frac{1}{N_c}\sum_{n=1}^{N_c}\frac{1}{HW}\sum_{i=1,j=i}^{H,W}\frac{\partial E(n)}{\partial x_{ij}^{lk}(n)} \tag{2}$$

where H and W are the spatial dimensions of layer $l$ and $N_c$ is the total number of images from the category (here $N_C = 35$, and the merged images used were generated from the same images used to generate tuning curves, described below). $E(n)$ is the error of the 1000-way classifier in response to image $n$, which is defined as the cross entropy between the activity vector of the final layer (after the soft-max operation) and a one-hot vector, wherein the correct label is the only non-zero entry. Because we are interested in activity changes that would decrease the error value, we negate this term. The gradient value we end up with thus indicates how the feature map's activity would need to change to make the network more likely to classify an image as the desired category. Repeating this procedure for each category, we obtain a set of gradient values (one for each category, akin to a tuning curve), for each feature map: $g^{lk}$. Note that, as these values result from applying the chain rule through layers of the network, they can be very small, especially for the earliest layers. For this study, the sign and relative magnitudes are of more interest than the absolute values.

## Oriented grating attention tasks

In addition to attending to object categories, we also test attention on simpler stimuli. In the orientation detection task, the network detects the presence of a given orientation in an image. Again, the final layer of the network is replaced by a series of binary classifiers, one for each of 9 orientations (0, 20, 40, 60, 80, 100, 120, 140, and 160 degrees. Gratings had a frequency of. 025 cycles/pixel). The training sets for each were balanced (50% had only the given orientation and 50% had one of 8 other orientations) and composed of full field (224 by 224 pixel) oriented gratings in red, blue, green, orange, or purple (to increase the diversity of the training images, they were randomly degraded by setting blocks of pixels ranging uniformly from 0% to 70% of the image to 0 at

random). Test images were each composed of two oriented gratings of different orientation and color (same options as training images). Each of these gratings were of size 112 by 112 pixels and placed randomly in a quadrant while the remaining two quadrants were black (*Figure 5A*). Again, the test sets were balanced and performance was measured as the average of the true positive and true negative rates (100 test images per orientation).

These same test images were used for a task wherein the network had to classify the color of the grating that had the attended orientation (cross-featural task paradigms like this are commonly used in attention studies, such as *Sàenz et al., 2003*). For this, the final layer of the network was replaced with a 5-way softmax color classifier. This color classifier was trained using the same full field oriented gratings used to train the binary classifiers (therefore, the network saw each color at all orientation values).

For another analysis, a joint feature and spatial attention task was used. This task is almost identical to the setup of the orientation detection task, except that the searched-for orientation would only appear in one of the four quadrants. Therefore, performance could be measured when applying feature-based attention to the searched-for orientation, spatial attention to the quadrant in which it could appear, or both.

## How attention is applied

This study aims to test variations of the feature similarity gain model of attention, wherein neural activity is modulated by attention according to how much the neuron prefers the attended stimulus. To replicate this in our model, we therefore must first determine the extent to which units in the network prefer different stimuli ('tuning values'). When attention is applied to a given category, for example, units' activities are modulated according to these values.

### Tuning values

To determine tuning to the 20 object categories used, we presented the network with images of each object category (the same images on which the binary classifiers were trained) and measured the relative activity levels. Because feature-based attention is a spatially global phenomenon (*Zhang and Luck, 2009*; *Saenz et al., 2002*), we treat all units in a feature map identically, and calculate tuning by averaging over them.

Specifically, for the $k^{th}$ feature map in the $l^{th}$ layer, we define $r^{lk}(n)$ as the activity in response to image $n$, averaged over all units in the feature map (i.e., over the spatial dimensions). Averaging these values over all images in the training sets ($N_c = 35$ images per category, 20 categories. $N = 700$) gives the mean activity of the feature map $\bar{r}^{lk}$:

$$\bar{r}^{lk} = \frac{1}{N} \sum_{n=1}^{N} r^{lk}(n) \tag{3}$$

Tuning values are defined for each object category, $c$ as:

$$f_c^{lk} = \frac{\frac{1}{N_c} \sum_{n \in c} r^{lk}(n) - \bar{r}^{lk}}{\sqrt{\frac{1}{N} \sum_{n=1}^{N} \left( r^{lk}(n) - \bar{r}^{lk} \right)^2}} \tag{4}$$

That is, a feature map's tuning value for a given category is merely the average activity of that feature map in response to images of that category, with the mean activity under all image categories subtracted, divided by the standard deviation of the activity across all images. These tuning values determine how the feature map is modulated when attention is applied to the category. Taking these values as a vector over all categories, $\mathbf{f}_{lk}$, gives a tuning curve for the feature map. We define the overall tuning quality of a feature map as its maximum absolute tuning value: $max(|\mathbf{f}_{lk}|)$. To determine expected tuning quality by chance, we shuffled the responses to individual images across category and feature map at a given layer and calculated tuning quality for this shuffled data.

We also define the category with the highest tuning value as that feature map's most preferred, and the category with the lowest (most negative) value as the least or anti-preferred.

We apply the same procedure to generate tuning curves for orientation by using the full field gratings used to train the orientation detection classifiers. The orientation tuning values were used when applying attention in these tasks.

When measuring how correlated tuning values are with gradient values, shuffled comparisons are used. To do this shuffling, correlation coefficients are calculated from pairing each feature map's tuning values with a random other feature map's gradient values.

### Gradient values

In addition to applying attention according to tuning, we also attempt to generate the 'best possible' attentional modulation by utilizing gradient values. These gradient values are calculated slightly differently from those described above ('Object category gradient calculations'), because they are meant to represent how feature map activity should change in order to increase binary classification performance, rather than just increase the chance of classifying an image as a certain object.

The error functions used to calculate gradient values for the category and orientation detection tasks were for the binary classifiers associated with each object/orientation. A balanced set of test images was used. Therefore a feature map's gradient value for a given object/orientation is the averaged activity change that would increase binary classification performance for that object/orientation. Note that on images that the network already classifies correctly, gradients are zero. Therefore, the gradient values are driven by the errors: false negatives (classifying an image as not containing the category when it does) and false positives (classifying an image as containing the category when it does not). In our detection tasks, the former error is more prevalent than the latter, and thus is the dominant impact on the gradient values. Because of this, gradient values calculated this way end up very similar to those described in Materials and methods, 'Object category gradient calculations', as they are driven by a push to positively classify the input as the given category.

The same procedure was used to generate gradient values for the color classification task. Here, gradients were calculated using the 5-way color classifier: for a given orientation, the color of that orientation in the test image was used as the correct label, and gradients were calculated that would lead to the network correctly classifying the color. Averaging over many images of different colors gives one value per orientation that represents how a feature map's activity should change in order to make the network better at classifying the color of that orientation.

In the orientation detection task, the test images used for gradient calculations (50 images per orientation) differed from those used to assess performance. For the object detection task, images used for gradient calculations (45 per category; preliminary tests for some categories using 90 images gave similar results) were drawn from the same pool as, but different from, those used to test detection performance. Gradient values were calculated separately for merged and array images.

### Spatial attention

In the feature similarity gain model of attention, attention is applied according to how much a cell prefers the attended feature, and location is considered a feature like any other. In CNNs, each feature map results from applying the same filter at different spatial locations. Therefore, the 2-D position of a unit in a feature map represents more or less the spatial location to which that unit responds. Via the max-pooling layers, the size of each feature map shrinks deeper in the network, and each unit responds to a larger area of image space, but the 'retinotopy' is still preserved. Thus, when we apply spatial attention to a given area of the image, we enhance the activity of units in that area of the feature maps and decrease the activity of units in other areas. In this study, spatial attention is applied to a given quadrant of the image.

### Implementation options

The values discussed above determine how strongly different feature maps or units should be modulated under different attentional conditions. We will now lay out the different implementation options for that modulation. The multiplicative bidirectional form of attention is used throughout this paper (with the exception of *Figure 3—figure supplement 2* where it is compared to the others). Other implementations are only used for the Supplementary Results.

First, the modulation can be multiplicative or additive. That is, when attending to category $c$, the slope of the rectified linear units can be multiplied by the tuning value for category $c$ weighted by the strength parameter, $\beta$:

$$x_{ij}^{lk} = \left(1 + \beta f_c^{lk}\right)\left[\left(I_{ij}^{lk}\right)\right]_+ \tag{5}$$

with $I_{ij}^{lk}$ representing input to the unit coming from layer $l-1$. Alternatively, a weighted version of the tuning value can be added before the rectified linear unit:

$$x_{ij}^{lk} = \left[I_{ij}^{lk} + \mu_l \beta f_c^{lk}\right]_+ \tag{6}$$

Strength of attention is varied via the strength parameter, $\beta$. For the additive effect, manipulations are multiplied by $\mu_l$, the average activity level across all units of layer $l$ in response to all images (for each of the 13 layers respectively: 20, 100, 150, 150, 240, 240, 150, 150, 80, 20, 20, 10, 1). When gradient values are used in place of tuning values, we normalize them by the maximum value at a layer, to be the same order of magnitude as the tuning values: $\mathbf{g}^l/max(|g^l|)$.

Recall that for feature-based attention all units in a feature map are modulated the same way, as feature-based attention has been found to be spatially global. In the case of spatial attention, however, tuning values are not used and a unit's modulation is dependent on its location in the feature map. Specifically, the tuning value term is set to +1 if the $i,j$ position of the unit is in the attended quadrant and to $-1$ otherwise. For feature-based attention tasks, $\beta$ ranged from 0 to a maximum of 11.85 (object attention) and 0 to 4.8 (orientation attention). For spatial attention tasks, it ranged from 0 to 1.

Next, we chose whether attention only enhances units that prefer the attended feature, or also decreases activity of those that don't prefer it. For the latter, the tuning values are used as-is. For the former, the tuning values are positively-rectified: $\left[\mathbf{f}^{lk}\right]_+$.

Combining these two factors, there are four implementation options: additive positive-only, multiplicative positive-only, additive bidirectional, and multiplicative bidirectional.

The final option is the layer in the network at which attention is applied. We try attention at all convolutional layers individually and, in *Figure 3—figure supplement 1*, simultaneously (when applying simultaneously the strength range tested is a tenth of that when applying to a single layer).

## Signal detection calculations

For the joint spatial-feature attention task (*Figure 5*), we calculated criteria ($c$, 'threshold') and sensitivity ($d'$) using true (TP) and false (FP) positive rates as follows (*Luo and Maunsell, 2015*):

$$c = -0.5\left(\Phi^{-1}(TP) + \Phi^{-1}(FP)\right) \tag{7}$$

where $\Phi^{-1}$ is the inverse cumulative normal distribution function. $c$ is a measure of the distance from a neutral threshold situated between the mean of the true negative and true positive distributions. Thus, a positive $c$ indicates a stricter threshold (fewer inputs classified as positive) and a negative $c$ indicates a more lenient threshold (more inputs classified as positive). The sensitivity was calculated as:

$$d' = \Phi^{-1}(TP) - \Phi^{-1}(FP) \tag{8}$$

This measures the distance between the means of the distributions for true negative and two positives. Thus, a larger $d'$ indicates better sensitivity.

To prevent the individual terms in these expressions from going to $\pm\infty$, false positive rates of <.01 were set to .01 and true positive rates of >.99 were set to .99.

## Assessment of feature similarity gain model and feature matching behaviour

In *Figure 6*, we examined the effects that applying attention at certain layers in the network (specifically 2, 6, 8, 10, and 12) has on activity of units at other layers. Attention was applied with $\beta = .5$.

The recording setup is designed to mimic the analysis of (*Martinez-Trujillo and Treue, 2004*). Here, the images presented to the network are full-field oriented gratings of all orientation-color combinations. Feature map activity is measured as the spatially averaged activity of all units in a feature map in response to an image. Activity in response to a given orientation is further averaged

over all colors. We calculate the ratio of activity when attention is applied to a given orientation (and the orientation is present in the image) over activity in response to the same image when no attention is applied. These ratios are then organized according to orientation preference: the most preferred is at location 0, then the average of next two most preferred at location 1, and so on with the average of the two least preferred orientations at location 4 (the reason for averaging of pairs is to match *Martinez-Trujillo and Treue, 2004* as closely as possible). Fitting a line to these points gives a slope and intercept for each feature map (lines are fit using the least squares method). FSGM predicts a negative slope and an intercept greater than one.

To test for signs of feature matching behaviour, each feature map's preferred (most positive tuning value) and anti-preferred (most negative tuning value) orientations are determined. Activity is recorded when attention is applied to the preferred or anti-preferred orientation and activity ratios are calculated. According to the FSGM, activity when the preferred orientation is attended should be greater than when the anti-preferred is attended, regardless of whether the image is of the preferred or anti-preferred orientation. According to the feature matching (FM) model, however, activity when attending the presented orientation should be greater than activity when attending an absent orientation, regardless of whether the orientation is preferred or not. Therefore, we say that a feature map is displaying feature matching behaviour if (1) activity is greater when attending the preferred orientation when the preferred is present versus when the anti-preferred is present, and (2) activity is greater when attending the anti-preferred orientation when the anti-preferred is present versus when the preferred is present. The second criteria distinguishes feature matching behaviour from FSGM.

## Correlating activity changes with performance

In *Figure 6—figure supplement 2*, we use two different measures of attention-induced activity changes in order to probe the relationship between activity and classification performance. In both cases, the network is performing the orientation detection task described in *Figure 5A* and performance is measured only in terms of true positive rates. Because we know attention to increase both true and false positive rates, we would expect a positive correlation between activity changes and true positive performance, but a negative correlation between activity changes and true negative rates. This predicts that activity changes will have a monotonic relationship with true positive performance, but an inverted U-shaped relationship with total performance. Since we are calculating correlation coefficients of activity with performance, which measure a linear relationship, we use the rate of true positives as our measure of performance.

The first measure is meant to capture feature similarity gain model-like behaviour in a way similar to the metric described in *Figure 6B*. The main difference is that that measure is calculated over a population of images of different stimuli, whereas the variant introduced here can be calculated on an image-by-image basis. Images that contain a given orientation are shown to the network and the spatially-averaged activity of feature maps is recorded when attention is applied to that orientation and when it is not. The ratio of these activities is then plotted against each feature map's tuning value for the orientation. According to the FSGM, this ratio should be above one for feature maps with positive tuning values and less than one for those with negative tuning values. Therefore, we use the slope of the line fitted to these ratios plotted as a function of tuning values as an indication of the extent to which activity is FSGM-like (with positive slopes more FSGM-like). The median slope over a set of images of a given orientation is paired with the change in performance on those images with attention. This gives one pair for each combination of orientation, strength ($\beta = .15, .30, .45, .60, .75, .90$), and layer at which attention was applied (activity changes are only recorded if attention was applied at or before the recorded layer). The correlation coefficient between these value pairs is plotted as the orange line in *Figure 6—figure supplement 2C*.

The second measure aims to characterize activity in terms of its downstream effects, rather than the contents of the input ('Vector Angle' measure, see *Figure 6—figure supplement 2A* for a visualization). It is therefore more aligned with the gradient-based approach to attention rather than tuning, and is thus related to 'choice probability' measures (*Zaidel et al., 2017*; *Purushothaman and Bradley, 2005*). First, for a particular orientation, images that both do and do not contain that orientation are shown to the network. Activity (spatially-averaged over each feature map) in response to images classified as containing the orientation (i.e., both true and false positives) is averaged in order to construct a vector in activity space that represents positive classification for a given layer.

To reduce complications of working with vectors in high dimensions, principal components are found that capture at least 90% of the variance of the activity in response to all images, and all computations are done in this lower dimensional space. The next step is to determine if attention moves activity in a given layer closer to this direction of positive classification. For this, only images that contain the given orientation are used. For each image, the cosine of the angle between the positive-classification vector and the activity in response to the image is calculated. The median of these angles over a set of images is calculated separately for when attention is applied and when it is not. The difference between these medians (with-attention minus without-attention) is paired with the change in performance that comes with attention on those images. Then the same correlation calculation is done with these pairs as described above.

The outcome of these analyses is a correlation coefficient between the measure of activity changes and performance changes. This gives two values per layer: one for the FSGM-like measure and one for the vector angle measure. To determine if these two values are significantly different, we performed a bootstrap analysis. For this, correlation coefficients were recalculated using simulated data made by sampling with replacement from the true data. We do this 100 times and perform a two-sided t-test to test for differences between the two measures.

## Experimental data

Model results were compared to previously published data coming from several studies. In *Lupyan and Ward, 2013*, a category detection task was performed using stereogram stimuli (on object present trials, the object image was presented to one eye and a noise mask to another). The presentation of the visual stimuli was preceded by a verbal cue that indicated the object category that would later be queried (cued trials) or by meaningless noise (uncued trials). After visual stimulus presentation, subjects were asked if an object was present and, if so, if the object was from the cued category (categories were randomized for uncued trials). In Experiment 1 ('Cat-Drawings' in *Figure 4B*), the object images were line drawings (one per category) and the stimuli were presented for 1.5 s. In Experiment 2 ('Cat-Images'), the object images were grayscale photographs (multiple per category) and presented for 6 s (of note: this presumably allows for several rounds of feedback processing, in contrast to our purelfeed forwardrd model). True positives were counted as trials wherein a given object category was present and the subject correctly indicated its presence when queried. False positives were trials wherein no category was present and subjects indicated that the queried category was present.

In *Lupyan and Spivey (2010)*, a similar detection task was used. Here, subjects detected the presence of an uppercase letter that (on target present trials) was presented rapidly and followed by a mask. Prior to the visual stimulus, a visual ('Letter-Vis') or audio ('Letter-Aud') cue indicated a target letter. After the visual stimulus, the subjects were required to indicate whether any letter was present. True positives were trials in which a letter was present and the subject indicated it (only uncued trials or validly cued trials—where the cued letter was the letter shown—were considered here). False positives were trials where no letter was present and the subject indicated that one was.

The task in *Koivisto and Kahila (2017)* was also an object category detection task ('Objects'). Here, an array of several images was flashed on the screen with one image marked as the target. All images were color photographs of objects in natural scenes. In certain blocks, the subjects knew in advance which category they would later be queried about (cued trials). On other trials, the queried category was only revealed after the visual stimulus (uncued). True positives were trials in which the subject indicated the presence of the queried category when it did exist in the target image. False positives were trials in which the subject indicated the presence of the cued category when it was not in the target image. Data from trials using basic category levels with masks were used for this study.

Finally, we include one study using macaques ('Ori-Change') wherein both neural and performance changes were measured (*Mayo and Maunsell, 2016*). In this task, subjects had to report a change in orientation that could occur in one of two stimuli. On cued trials, the change occurred in the cued stimulus in 80% of trials and the uncued stimulus in 20% of trials. On neutrally-cued trials, subjects were not given prior information about where the change was likely to occur (50% at each stimulus). Therefore performance could be compared under conditions of low (uncued stimuli), medium (neutrally cued stimuli), and high (cued stimuli) attention strength. Correct detection of an orientation change in a given stimulus (indicated by a saccade) is considered a true positive and a

saccade to the stimulus prior to any orientation change is considered a false positive. True negatives are defined as correct detection of a change in the uncued stimulus (as this means the subject correctly did not perceive a change in the stimulus under consideration) and false negatives correspond to a lack of response to an orientation change. While this task includes a spatial attention component, it is still useful as a test of feature-based attention effects. Previous work has demonstrated that, during a change detection task, feature-based attention is deployed to the pre-change features of a stimulus (*Cohen and Maunsell, 2011*; *Mayo et al., 2015*). Therefore, because the pre-change stimuli are of differing orientations, the cueing paradigm used here controls the strength of attention to orientation as well.

In cases where the true and false positive rates were not published, they were obtained via personal communications with the authors. Not all changes in performance were statistically significant, but we plot them to show general trends.

We calculate the activity changes required in the model to achieve the behavioural changes observed experimentally by using the data plotted in *Figure 4B*. We determine the average $\beta$ value for the neutral and cued conditions by finding the $\beta$ value of the point on the model line nearest to the given data point. Specifically, we average the $\beta$ values found for the four datasets whose experiments are most similar to our merged image task (Cat-Drawings, Cat-Images, Letter-Aud, and Letter-Vis).

## Acknowledgements

We are very grateful to the authors who so readily shared details of their behavioural data upon request: J Patrick Mayo, Gary Lupyan, and Mika Koivisto. We further thank J Patrick Mayo for helpful comments on the manuscript.

## Additional information

### Funding

| Funder | Grant reference number | Author |
| --- | --- | --- |
| National Science Foundation | DBI-1707398 | Kenneth D Miller |
| National Institutes of Health | T32 NS064929 | Kenneth D Miller |
| Gatsby Charitable Foundation | | Kenneth D Miller |
| Google | | Grace W Lindsay |
| National Science Foundation | IIS-1704938 | Kenneth D Miller |

The funders had no role in study design, data collection and interpretation, or the decision to submit the work for publication.

### Author contributions

Grace W Lindsay, Conceptualization, Funding acquisition, Investigation, Visualization, Methodology, Writing—original draft; Kenneth D Miller, Supervision, Funding acquisition, Methodology, Writing—review and editing

### Author ORCIDs

Grace W Lindsay (iD) http://orcid.org/0000-0001-9904-7471
Kenneth D Miller (iD) http://orcid.org/0000-0002-1433-0647

### Decision letter and Author response

Decision letter https://doi.org/10.7554/eLife.38105.029
Author response https://doi.org/10.7554/eLife.38105.030

## Additional files

### Supplementary files

• Transparent reporting form
DOI: https://doi.org/10.7554/eLife.38105.023

### Data availability

The weights for the model used are linked to in the study. The data resulting from simulations have been packaged and are available on Dryad (doi:10.5061/dryad.jc14081). The analysis code are available on GitHub (https://github.com/gwl2108/CNN_attention; copy archived at https://github.com/elifesciences-publications/CNN_attention).

The following dataset was generated:

| Author(s) | Year | Dataset title | Dataset URL | Database and Identifier |
|---|---|---|---|---|
| Lindsay G, Miller K | 2018 | Data from: How biological attention mechanisms improve task performance in a large-scale visual system model | https://dx.doi.org/10.5061/dryad.jc14081 | Available at Dryad Digital Repository under a CC0 Public Domain Dedication, 10.5061/dryad.jc14081 |

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
