## [Decision Letter]

Thank you for submitting your article "How biological attention mechanisms improve task performance in a large-scale visual system model" for consideration by *eLife*. Your article has been reviewed by three peer reviewers, including Marcel van Gerven as the Reviewing Editor, and the evaluation has been overseen by Sabine Kastner as the Senior Editor. The following individual involved in review of your submission has agreed to reveal his identity: Marius Peelen (Reviewer #1).

The reviewers have discussed the reviews with one another and the Reviewing Editor has drafted this decision to help you prepare a revised submission.

Summary:

The authors present an approach to examining the feature similarity model of attention by incorporating attentional modulation in a convolutional neural network for object categorization. They find that task performance enhancements with attention can roughly approximate those found experimentally, but most interestingly, only when attention is applied to the later layers of the network. The authors demonstrate a dissociation between layers in the strength of tuning and the performance enhancement achieved by applying attention. This represents an interesting contribution to theories of attention to the extent that one considers CNNs a useful model for biological vision. Another key contribution is the distinction between gradient-based and tuning-based feedback. The findings obtained here in neural networks make several new predictions for biological experiments and raise fundamental questions about how tuning affects behavior.

Essential revisions:

Introduction, second paragraph: CNNs are introduced as great models of the ventral stream, but an increasing number of studies shows that the features used in these CNNs to classify objects are very different from those used by humans (e.g., Azulay and Weiss, 2018; Baker, Lu, Erlikhman, and Kelleman VSS 2018; Ullman et al., 2016). Some caution may be warranted.

In Figure 3 performance increase across layers is shown. This plot is created by using the best performing weighting parameter *β*. To dissociate the effect of *β* and of the attentional modulation f, please add a control condition in which f is set to one. The reasoning is that varying *β* alone and picking the best *β* may already induce performance changes.

In Figure 3 the results for modulating all layers are shown. It is felt that the conclusions drawn from these results are unsupported. Picking *β* at 1/10 of the optimal *β* for each layer does not constitute an optimal setting for modulating all layers. Also, modulations in early layers may negatively impact activity changes in later layers. Hence, the authors cannot exclude the possibility that modulation of all layers simultaneously could actually help. Results and interpretations of attention applied to all layers should therefore be removed from the paper.

In the subsection “Attention Strength and the Tradeoff between Increasing True and False Positives”, you compare the change in the magnitude of neural activation in the CNN to the changes in primate brains. It was not clear how to interpret these results. Can these magnitudes be meaningfully compared? What can we conclude from this?

In the subsection “Feature-based Attention Primarily Influences Criteria and Spatial Attention Primarily Influences Sensitivity”, it is argued that FBA works through a criterion shift rather than by increasing sensitivity, with FBA shifting the representation of all stimuli in the direction of the attended category. But earlier you show that FBA selectively increases TP (relative to FP), which suggests an increase in sensitivity. (Also, Figure 4E appears to show a positive effect of FBA (L13) on sensitivity). Please clarify.

For many of the analyses results of both types of feedback are shown. However, for some comparisons only tuning-based results are shown (e.g., see the aforementioned subsection). Why? Please ensure consistency throughout.

It is unclear why a new method for quantifying attention is introduced in Figure 7, or how the "FSGM-like" measure is related to feature matching and the activity ratios already discussed. Please motivate or restrict to feature matching and activity ratios. In general, the paper is a dense read due to the various analysis and metrics. Any steps towards simplification of the presentation will aid the reader.

The claim that the new measure of attention (Figure 7A), or the alternative measures of attention for that matter, is experimentally testable seems unsupported. In particular, getting with and without attention activity in response to images that are not classified as the target orientation is not possible to measure in experiments with humans or animals. Subjects are stochastic in their judgments. One could however, measure with and without attention responses to ambiguous stimuli that elicit near chance performance. This metric would then become very similar to a population version of the well-studied "choice probability" metric. This connection should at least be discussed.

It would be more useful to the experimental community to recast the orientation task and analysis more in terms of what would be measured empirically. For instance presenting the task in terms of correctly identified target orientation as a function of the presented orientations rotation from the target. This may be outside the scope of current manuscript though.

What would be the biological mechanism that can account for tuning-based and gradient-based feedback? Especially the gradient based approach seems to be hard to defend from a biological point of view. How would putative decision-related areas have access to this gradient information? Some words should be spent on this in the Discussion section.

Please mention relevant related work:

- Katz et al., 2016, related to the relationship between tuning and influence on decisions.

- Abdelhack and Kamitani, 2018, related to subsection “Recordings Show How Feature Similarity Gain Effects Propagate” (and Figure 7) showing that the activity in response to misclassified stimuli shifts towards the activity in response to correctly classified stimuli when attention is turned on.

- Stein and Peelen, 2015, related to the subsection “Feature-based Attention Primarily Influences Criteria and Spatial Attention Primarily Influences Sensitivity”, arguing that FBA in human experiments does not lead to an increase in sensitivity (see also work by Carrasco on effects of FBA on discrimination tasks).

- Discussion, fifth paragraph: Ni, Ray, and Maunsell, 2012, would appear to be very relevant to this Discussion section. Those authors found that strength of normalization was as strong a factor as tuning in the strength of attentional effects.

- The neural network community developed various models that implement some form of attention. See the Attention section in Hassabis et al., Neuron, 2017. The present work should be contrasted with the papers mentioned there.

---

## [Author Response]

Essential revisions:Introduction, second paragraph: CNNs are introduced as great models of the ventral stream, but an increasing number of studies shows that the features used in these CNNs to classify objects are very different from those used by humans (e.g., Azulay and Weiss, 2018; Baker, Lu, Erlikhman, and Kelleman VSS 2018; Ullman et al., 2016). Some caution may be warranted.

The language and references have been updated to reflect this (Introduction, second paragraph).

In Figure 3 performance increase across layers is shown. This plot is created by using the best performing weighting parameter β. To dissociate the effect of β and of the attentional modulation f, please add a control condition in which f is set to one. The reasoning is that varying β alone and picking the best β may already induce performance changes.

This control has been added as Figure 3—figure supplement 1C, subsection “Feature-based Attention Improves Performance on Challenging Object Classification Tasks”, fourth paragraph.

In Figure 3 the results for modulating all layers are shown. It is felt that the conclusions drawn from these results are unsupported. Picking β at 1/10 of the optimal β for each layer does not constitute an optimal setting for modulating all layers. Also, modulations in early layers may negatively impact activity changes in later layers. Hence, the authors cannot exclude the possibility that modulation of all layers simultaneously could actually help. Results and interpretations of attention applied to all layers should therefore be removed from the paper.

We shifted the all-layer results to supplementary figures. Although it is true that we have not exhaustively explored the possibilities with modulation at all layers, we believe that the particular case we did study is of sufficient interest to readers to be included in the supplement. What we have shown is that modulating all layers, with 1/10 the strength of a single layer, improves performance about as much as modulation of the best single layer. Figure 3—figure supplements 1A and B, subsection “Feature-based Attention Improves Performance on Challenging Object Classification Tasks”, fourth paragraph.

In the subsection “Attention Strength and the Tradeoff between Increasing True and False Positives”, you compare the change in the magnitude of neural activation in the CNN to the changes in primate brains. It was not clear how to interpret these results. Can these magnitudes be meaningfully compared? What can we conclude from this?

We feel that the main take-away from that calculation is that the neural changes needed to cause observed performance chances are neither absurdly large nor absurdly small when compared to the what is observed experimentally in neurons. Therefore, we believe that as an order-of-magnitude comparison it provides some insight and reassurance that the model may be working similarly to the biology. We have shortened this section to make it clearer that this is the intended interpretation (subsection “Attention Strength and the Tradeoff between Increasing True and False Positives”, eighth paragraph).

In the subsection “Feature-based Attention Primarily Influences Criteria and Spatial Attention Primarily Influences Sensitivity”, it is argued that FBA works through a criterion shift rather than by increasing sensitivity, with FBA shifting the representation of all stimuli in the direction of the attended category. But earlier you show that FBA selectively increases TP (relative to FP), which suggests an increase in sensitivity. (Also, Figure 4E appears to show a positive effect of FBA (L13) on sensitivity). Please clarify.

We’d like to make two points of clarification to the reviewers on this regard:

First, the claim we make about feature-based attention (FBA) vs. spatial attention is that FBA has a larger impact on criteria than sensitivity and vice versa for spatial attention. Thus, we do not claim that there are no sensitivity changes coming from FBA, only that the criteria changes are more dominant.

Second, the fact that FBA first (with increasing modulation strength) increases TP (true positives) much more than FP (false positives), and only later (for much stronger modulation strength) increases FP more than TP, need not suggest an increase in sensitivity (where increasing sensitivity means increasing the separation between the distribution of TP and FP). This outcome could result solely from a criteria change (a decrease in threshold), as shown by the example in Author response image 1.

Here, the shaded area representing true positives will increase as the threshold moves leftward, before the false positives increase.

The text reflects these points (subsection “Feature-based Attention Primarily Influences Criteria and Spatial Attention Primarily Influences Sensitivity”, second paragraph).

For many of the analyses results of both types of feedback are shown. However, for some comparisons only tuning-based results are shown (e.g., see the aforementioned subsection). Why? Please ensure consistency throughout.

The gradient-based results were omitted in Figure 5E only due to the already crowded nature of that graph. Gradient results have now been added to that figure.

It is unclear why a new method for quantifying attention is introduced in Figure 7, or how the "FSGM-like" measure is related to feature matching and the activity ratios already discussed. Please motivate or restrict to feature matching and activity ratios. In general, the paper is a dense read due to the various analysis and metrics. Any steps towards simplification of the presentation will aid the reader.

The feature-similarity gain model (FSGM) measure in Figure 6 was taken to match a measure used in previous experimental work. It requires the presentation of multiple orientations in order to calculate the impact of attention on a given cell. That is, it measures how the effects of attention change across stimuli for a single cell. In Figure 7, we wanted to test for FSGM-like activity changes for a single stimulus, because this allows us to best study the correlation of FSGM-like activity changes with performance changes (because performance varies from stimulus to stimulus). Therefore we had to measure how the effects of attention change across cells, i.e. across the population as a whole, for responses to a single orientation (or even a single image). That is why we introduce a new “FSGM-like” measure in Figure 7. Both measures test whether attention multiplicatively increases (decreases) the activity of cells that are tuned to prefer (anti-prefer) the attended stimulus. The difference is simply whether the attention vs. no-attention activity ratios used to fit the line are taken from the same cell (which for our model means the same feature map) in response to different orientations (Figure 6B), or from all the feature maps in response to the same orientation (Figure 7). Our new population measure, like the single-cell measure which was used previously in experiments, can be easily implemented by experimentalists. We agree with the reviewers that we did not make our reasons for using a new measure, and the precise difference between the two measures, clear, and we have now made these points clear in our revision.

Having a measure of FSGM-like activity that could be calculated per-orientation allowed us to make a more fine-grained analysis of the correlation between FSGM-like activity changes and performance changes (because we did not have to average performance over all orientations). We show these correlations in Figure 7C. In that figure we also introduced a second measure of attention-induced activity changes (vector angle). This measure was introduced to see if a in hopes of finding a different measure of activity changes that better correlates with performance changes than the FSGM measure. The vector angle measure was meant to be an experimentally tractable measure that better reflects gradients rather than FSGM-like tuning curves, following our findings that the gradients should be most potent in affecting performance. We show that, at early layers, the vector angle measure of attention-induced activity changes does indeed better correlate with performance changes than the FSGM-like measure.

However, we agree with the reviewers that Figure 7 and the associated new analyses are complex and could be confusing. Yet we believe the proposal of different measures of activity changes and the exploration of their correlation with performance changes are valuable contributions from this kind of modeling work. Furthermore the results of Figure 7 were modest rather than striking. We therefore referenced the findings of these analyses in the main text and moved Figure 7, and a detailed explanation of the different measures and their motivations, to the supplementary materials.

In its place, we have included, as Figure 7, a figure illustrating the point we think will be of most interest to experimentalists at this point in our paper, namely a figure illustrating an experiment to clearly distinguish whether attention is applied according to gradients rather than (as always assumed up to this point) according to tuning. We previously only addressed this experiment in the Discussion, but we believe it is best to make it a prominent and illustrated part of the Results.

This new figure is now Figure 7, and the old Figure 7 has been moved to Figure 6—figure supplement 2. The analysis techniques in Figure 6—figure supplement 2 are referenced in the sixth paragraph of the subsection “Recordings Show How Feature Similarity Gain Effects Propagate”, and further explanation has been added in the Materials and methods subsection “Assessment of Feature Similarity Gain Model and Feature Matching Behavior” as well as to the caption of Figure 6—figure supplement 2. The experiment (Figure 7) is described in the last paragraph of the subsection “Recordings Show How Feature Similarity Gain Effects Propagate”.

The claim that the new measure of attention (Figure 7A), or the alternative measures of attention for that matter, is experimentally testable seems unsupported. In particular, getting with and without attention activity in response to images that are not classified as the target orientation is not possible to measure in experiments with humans or animals. Subjects are stochastic in their judgments. One could however, measure with and without attention responses to ambiguous stimuli that elicit near chance performance. This metric would then become very similar to a population version of the well-studied "choice probability" metric. This connection should at least be discussed.

The alternative measures (we take this to mean the FSGM measures of Figure 7 vs. Figure 6) have already, in the Figure 6 version, been used experimentally, and we see no reason why the population-based version of Figure 7 should not also be usable. We understand the rest of this comment to be referring to the “new measure”, namely the vector-angle method of assaying attention-induced activity changes. We agree with the reviewers that behavioral responses will be stochastic, whereas those in our model are deterministic, but disagree that this renders the vector-angle method unusable. As the reviewers suggest in referring to a threshold task, so long as the method is applied to images in which, without attention, there is a significant percentage of mis-categorizations, and attention improves performance, the measure will be usable: the method only requires that population responses be measured (1) without attention for mis-categorization (2) without attention for correct categorization and (3) with attention for both cases. We believe introducing this measure to experimentalists, which should better test the ability of attention to modulate gradients rather than tuning, will be a valuable contribution, although again, we have largely put this in the supplement.

Furthermore, it is not necessary for the blue and red vectors in Figure 7A to be calculated only using images that were negatively-classified without attention. The same result would be expected if all images were used. To be specific, a collection of images, some of which contained the desired orientation and some that did not, could be shown to a subject (who is not deploying attention to the orientation). The positive-classification vector (gray vector in 7A) would be made from the responses when an image was positively-classified (regardless of whether the desired orientation was present). The blue vector would be made from the responses to all images that contained the desired orientation (regardless of how they were classified), and the red vector would be from the responses to images that contained the desired orientation (regardless of how they were classified) but when attention was applied to that orientation. In this setting we would still expect the red vector (attention present) to be closer to positive classification vector than the blue (attention absent).

The results of calculating these vectors this way are shown in the green lines in Author response image 2 (with original analyses for comparison), and give results virtually identical to those of our previously-used definition of vector angle (blue lines).

**Author response image 2. respfig2:** 

Figure 6—figure supplement 2 now uses the new definition of the Vector Angle measure (that which uses all images, green line in Author response image 2). The last paragraph of the subsection “Assessment of Feature Similarity Gain Model and Feature Matching Behavior” has been changed to reflect that.

It would be more useful to the experimental community to recast the orientation task and analysis more in terms of what would be measured empirically. For instance presenting the task in terms of correctly identified target orientation as a function of the presented orientations rotation from the target. This may be outside the scope of current manuscript though.

We agree that relating the performance trends in these models more to behavioral data from various experimentally-used tasks would be interesting, but indeed outside the scope of the current manuscript.

What would be the biological mechanism that can account for tuning-based and gradient-based feedback? Especially the gradient based approach seems to be hard to defend from a biological point of view. How would putative decision-related areas have access to this gradient information? Some words should be spent on this in the Discussion section.

We believe there are means by which the gradient-based feedback could be learned biologically, through correlation-based or spike-timing-based learning rules in which, if lower-level activity is followed by high-level activity, the feedback from those higher-level units to the corresponding lower-level units is strengthened. We have fleshed this out in the Discussion (fifth paragraph).

Please mention relevant related work:- Katz et al., 2016, related to the relationship between tuning and influence on decisions.- Abdelhack and Kamitani, 2018, related to subsection “Recordings Show How Feature Similarity Gain Effects Propagate” (and Figure 7) showing that the activity in response to misclassified stimuli shifts towards the activity in response to correctly classified stimuli when attention is turned on.- Stein and Peelen, 2015, related to the subsection “Feature-based Attention Primarily Influences Criteria and Spatial Attention Primarily Influences Sensitivity”, arguing that FBA in human experiments does not lead to an increase in sensitivity (see also work by Carrasco on effects of FBA on discrimination tasks).- Discussion, fifth paragraph: Ni, Ray, and Maunsell, 2012, would appear to be very relevant to this Discussion section. Those authors found that strength of normalization was as strong a factor as tuning in the strength of attentional effects.- The neural network community developed various models that implement some form of attention. See the Attention section in Hassabis et al., Neuron, 2017. The present work should be contrasted with the papers mentioned there.

These citations have been included in the Discussion.